# Expression strategies for the efficient synthesis of antimicrobial peptides in plastids

Matthijs P. Hoelscher [1,2], Joachim Forner [1], Silvia Calderone [1,3], Carolin Krämer[1], Zachary Taylor [1], F. Vanessa Loiacono [1], Shreya Agrawal[1,4], Daniel Karcher[1], Fabio Moratti [1], Xenia Kroop[1] & Ralph Bock [1] ✉

Antimicrobial peptides (AMPs) kill microbes or inhibit their growth and are promising next-generation antibiotics. Harnessing their full potential as antimicrobial agents will require methods for cost-effective large-scale production and purification. Here, we explore the possibility to exploit the high protein synthesis capacity of the chloroplast to produce AMPs in plants. Generating a large series of 29 sets of transplastomic tobacco plants expressing nine different AMPs as fusion proteins, we show that high-level constitutive AMP expression results in deleterious plant phenotypes. However, by utilizing inducible expression and fusions to the cleavable carrier protein SUMO, the cytotoxic effects of AMPs and fused AMPs are alleviated and plants with wild-type-like phenotypes are obtained. Importantly, purified AMP fusion proteins display antimicrobial activity independently of proteolytic removal of the carrier. Our work provides expression strategies for the synthesis of toxic polypeptides in chloroplasts, and establishes transplastomic plants as efficient production platform for antimicrobial peptides.

Antimicrobial peptides (AMPs) are peptides that kill microbes or inhibit their growth[1–3]. AMPs represent an important part of defense strategies against pathogenic microbes, and are produced by both eukaryotic and prokaryotic life forms, although the majority of identified AMPs are from animals[4].

Due to the increasing prevalence of multidrug-resistant bacteria, AMPs have received considerable interest as an alternative to currently used antibiotics[5,6]. In addition to their potential use to treat infectious diseases in humans, AMPs are also considered as alternatives to antibiotics as growth promoters in animal husbandry[7].

AMPs are generally short polypeptides. The average length of all AMPs in the Antimicrobial Peptide Database (APD) is 33.19 amino acids[4] (database accessed in February 2022). The vast majority of AMPs have a positive net charge and are composed of a mix of hydrophobic and hydrophilic groups that fold into an amphipathic conformation, where one side of the molecule is hydrophobic and the other side hydrophilic[1]. The positive net charge of AMPs promotes electrostatic interactions with negatively charged bacterial membranes, but not with neutral eukaryotic membranes[1,8,9]. The hydrophobic regions of AMP molecules facilitate binding and integration into bacterial membranes[1,8]. Negatively charged and uncharged AMPs that kill bacteria also exist, although they are rare. Only 5.7% of bactericidal AMPs are uncharged and only 6% have a charge below zero[4] (APD accessed in February 2022). Since purification from the native host is usually not feasible and direct chemical synthesis is far too expensive (with the possible exception of some very small AMPs that do not require disulfide bridges or other post-translational modifications), recombinant expression using transgenic methods

[1]Max-Planck-Institut für Molekulare Pflanzenphysiologie, Am Mühlenberg 1, D-14476 Potsdam-Golm, Germany. [2]Present address: Utrecht University, Pharmaceutical sciences, Pharmaceutics, Universiteitsweg 99, 3584 CG Utrecht, Netherlands. [3]Present address: Centre for Research in Agricultural Genomics (CRAG), CSIC-IRTA-UAB-UB, Campus UAB, Bellaterra, 08193 Barcelona, Spain. [4]Present address: Neoplants, 630 Rue Noetzlin Bâtiment, 91190 Gif-sur-Yvette, France. ✉e-mail: rbock@mpimp-golm.mpg.de

represents the most promising strategy for commercial-level AMP production[10].

In spite of their bactericidal activity, recombinant production of some AMPs in bacteria is possible[11]. However, especially for the large-scale and cost-effective commercial production, plants represent the most promising production platform. Transgenic plants have been shown to facilitate the synthesis of large quantities of biopharmaceuticals at low costs, with the option to scale up production by simply expanding the planted area[12,13]. Both stably transformed and transiently transformed plants can be used for recombinant protein production. Transient expression offers the benefit that negative effects of transgene expression on growth can be avoided, and the response time for development of new therapeutics is short[13]. Stable expression systems rely on genetic transformation of either the nuclear or the plastid genome. For production purposes, they offer a number of significant advantages: The harvested biomass is free of transgenic pathogenic microbes (*Agrobacterium* or viruses), there is no requirement for transformation of every new batch of plant material (likely also resulting in much greater batch-to-batch consistency in protein accumulation levels) and substantially reducing cost[14], and facilitates scale-up to large greenhouse or open field cultivation[15]. The use of specialized promoter systems can trigger expression in all tissues, restrict transgene expression to specific tissues or enable inducible expression[16–21]. Expression from the plastid (chloroplast) genome is a particularly attractive method of heterologous protein production in plants, because chloroplasts can accumulate very large amounts of foreign protein[22–24], and offer the additional benefits of transgene exclusion from pollen transmission and the absence of epigenetic transgene silencing[25]. Production of AMPs or antimicrobial proteins in plants has been attempted by transient expression, stable nuclear transformation and plastid transformation[26–32].

When considering possibilities to develop strategies for the production of AMPs in plants by expression from the plastid genome (i.e., in transplastomic plants), several important considerations come into play. In view of the prokaryotic origin of plastids, a serious concern is that antimicrobial peptides may damage the plastid (and its membranes) in a similar manner as they kill bacteria. Problems with AMP production in bacteria have been extensively studied, and different solutions have been developed to enhance accumulation and reduce toxicity[10,33]. Recombinant AMP production in bacteria is commonly attempted by using an inducible expression system, to allow unhindered growth of the bacterial culture before induction of AMP synthesis[10,33]. In fact, inducible expression was key to the heterologous production of most of the AMPs that could be expressed in bacteria, indicating that the inherent toxicity of AMPs to prokaryotic production hosts represents a serious obstacle[33,34].

Another significant concern is that AMPs might be unstable inside plastids due to their small size, the presence of hydrophobic domains (which can be problematic in plastid transgene expression[35]). AMPs share these properties with the small plastid targeting peptides that are rapidly degraded after protein import has been completed[5,36,37].

In fact, AMPs and transit peptides for protein import into plastids and mitochondria are so similar that AMPs are considered as possible evolutionary ancestors of organellar targeting peptides[38]. The presequence protease (PreP) in plastids and mitochondria efficiently degrades cleaved-off transit peptides with a size of 10-65 amino acids[37], suggesting that polypeptides of similar size that are recombinantly expressed in plastids will also be degraded by PreP. PreP displays a preference for positively charged residues at the cleavage site[36], which could further contribute to degradation of positively charged (cationic) AMPs. In addition, plastids harbor an oligopeptidase (organellar oligopeptidase, OOP) that degrades peptides of 8-23 amino acids[36] and potentially also contributes to the instability of small polypeptides expressed in plastids. Notably, the chloroplast genome encodes a few small proteins, with PetN, a subunit of the cytochrome $b_6f$ complex of

only 29 amino acids[39], being the smallest example. However, all of these small proteins are subunits of protein complexes (photosynthetic complexes residing in the thylakoid membrane, chloroplast ribosomes), and the rapid (often co-translational) incorporation of these small proteins into large multiprotein complexes likely protects them from proteolytic degradation.

In this work, we systematically compare multiple approaches to maximize AMP expression levels in transplastomic tobacco plants, and simultaneously, reduce the toxic effects associated with high-level AMP accumulation in chloroplasts. Based on the above considerations, we hypothesize that embedding AMPs into larger polypeptides will increase their stability, because they will not be recognized as PreP and/or OPP substrates. In order to increase the size of AMPs, we pursue two parallel strategies: (i) the fusion of multiple antimicrobial peptides connected by flexible linkers, and (ii) the use of the small ubiquitin-like modifier (SUMO) as a cleavable fusion partner.

## Results

### Design of AMP fusion constructs for transplastomic expression

Given the above-described concerns related to AMP toxicity and instability due to their small size, we sought to test the idea that fusion proteins of several AMPs can be expressed to high levels in chloroplasts. In addition, combination of different AMPs could result in synergistic antimicrobial effects[40–42], thus providing more potent antimicrobials. In bacteria, recombinant production of AMPs is frequently performed as fusions to carrier proteins like glutathione S-transferase (GST), thioredoxin or small ubiquitin-like modifier (SUMO) – a strategy that increases AMP stability and solubility, and reduces toxicity to the host[10,33]. The carrier proteins can be removed after purification[33,43], and production of AMPs as SUMO fusions and subsequent cleavage and purification of the peptide were demonstrated in *E. coli*[44]. SUMO fusion and subsequent removal was also used to improve the production of the antiviral protein cyanovirin in *E. coli*[45]. Fusion proteins comprising multiple AMPs have been described in bacteria[33,41,42,46,47], but production in chloroplasts has not been attempted yet.

Nine AMPs were selected as promising candidates for the construction of fusion proteins and expression in tobacco chloroplasts (Supplementary Data 1). The AMPs cgMolluscidin, CXCL9 and ubiquicidin were selected because they possess a very strong positive charge, a property that could make them more selective towards negatively charged bacterial membranes[48–52] (Supplementary Data 1). They are also relatively large with 55, 103 and 59 amino acids, respectively (Supplementary Data 1). Six other AMPs, novispirin G10, esculentin-1-OA1, andersonin-Y1, wallaby antimicrobial 1 (WAM1), dermaseptin S4 (derivative $K_4K_{20}$-S4) and polyphemusin I were selected, because their minimal inhibitory concentrations (MIC) against both gram-positive and gram-negative bacteria were among the lowest to be found in the literature[4,53–58] (Supplementary Data 1).

ince plastids harbor proteases that specifically recognize and degrade small peptides between 10 and 65 amino acids[36,37], we anticipated that single AMPs would be prone to degradation. We, therefore, prepared single AMP expression constructs only for three of the selected AMPs: CXCL9 (with 103 amino acids the largest AMP included in our study and the only one above the 65 amino acid threshold; Supplementary Data 1), and HA epitope-tagged versions of novispirin and WAM1 (with 34 and 52 amino acids, respectively; Fig. 1a). In addition, to overcome the size-selective degradation, we designed a set of constructs for the expression of enlarged polypeptides by fusing multiple AMPs, separated by flexible linkers of 5 or 15 amino acids (Fig. 1a; Supplementary Table 1 and Supplementary Data 2). The use of flexible linkers can preserve the functionality of fused proteins, by allowing independent movement[59]. In this way, the fusion of different AMPs also offers the potential to yield synthetic antimicrobial polypeptides that are more potent than natural AMPs by synergistically combining different modes of action (Supplementary Data 1).

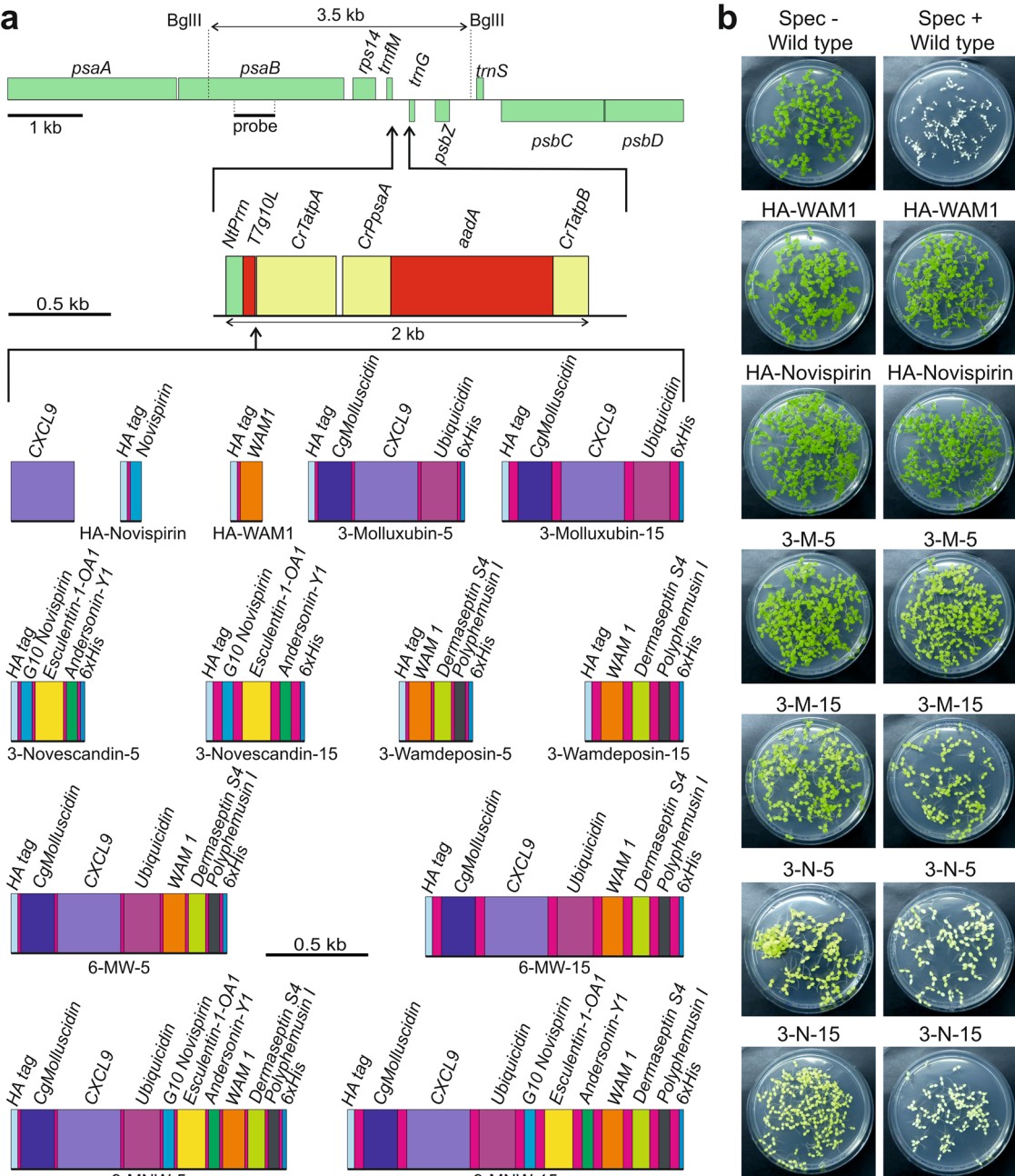

**Fig. 1 | Constitutive AMP and fAMP expression from the tobacco plastid genome. a** Sequences encoding AMPs or fAMPs were placed under the control of the *Nicotiana tabacum* plastid ribosomal RNA operon promoter (*NtPrrn*), the T7 phage *gene10* leader (*T7g10L*) as 5′ UTR and the 3′ UTR of the *atpA* gene from *Chlamydomonas reinhardtii (CrTatpA)*. The spectinomycin resistance gene *aadA* for selection of transplastomic lines was driven by the *Chlamydomonas psaA* promoter and 5′ UTR *(CrPpsaA)*, and the 3′ UTR of the *Chlamydomonas atpB* gene (*CrTatpB*). Different combinations of AMPs were linked with linkers of 5 or 15 amino acids (pink). The constructs were flanked by homology regions for integration into the

spacer between the *trnfM* and *trnG* genes in the tobacco plastid genome. Restriction sites (BglII) and the binding site of the probe for RFLP analysis to verify the transplastomic state are indicated. For information on AMPs and fAMPs, see Supplementary Tables 1, 2 and Supplementary Data 1 and 2. **b** Seed assays to confirm homoplasmy of transplastomic AMP-expressing lines. Seeds were sown on medium with (Spec+) or without (Spec−) spectinomycin, and photos were taken after two weeks. In the case of 3-Molluxubin-5 (3-M-5), 3-Molluxubin-15 (3-M-15) and 3-Novescandin-5 (3-N-5), seeds were obtained from grafted plants. 3-N-15: 3-Novescandin-15.

Expression constructs were generated in which combinations of three, six or nine different AMPs were fused with linkers of 5 or 15 amino acids (Fig. 1a; Supplementary Data 2). The resulting repeated use of linkers necessitated the systematic introduction of sequence variation at the DNA level (see Methods; Supplementary Table 1) to prevent construct instability in the chloroplast genome by homologous recombination on directly repeated identical sequences[60,61]. For the 15 amino acid linkers, sufficient DNA sequence variation could

only be achieved by additionally varying the encoded amino acid sequences (Supplementary Table 1).

To facilitate protein detection and purification, all fusions were equipped with N-terminal HA and C-terminal 6xHis tags. The combinations of three different antimicrobial peptides will subsequently be referred to as 3-Molluxubin (3-M), 3-Novescandin (3-N) and 3-Wamdeposin (3-W). The names are portmanteaus of the AMPs they are composed of (Supplementary Data 2), followed by the number 5 or

15 to distinguish the 5 and 15 amino acid linker versions (e.g., 3-Wam-deposin-5). The fusion of 6 AMPs was named 6-MW (combining 3-M and 3-W) and the fusion of all 9 AMPs was called 9-MNW (combining 3-M, 3-N and 3-W; Supplementary Data 2).

## Constitutive transplastomic expression of AMPs and AMP fusions

In an attempt to achieve maximal transgene expression levels, we initially utilized the strongest gene expression elements known for chloroplasts: the tobacco (Nicotiana tabacum) plastid ribosomal RNA operon promoter (Prrn) in combination with the gene10 leader from the coliphage T7[22,62] (Fig. 1a). The DNA sequences encoding the AMPs were codon optimized to match the preferred codon usage of tobacco chloroplasts[63].

The constructs were introduced into the tobacco plastid genome by particle bombardment (biolistic transformation) with DNA constructs containing the AMP transgenes inserted into a standard vector for chloroplast transformation[64] (Fig. 1a). The vector (pDK323) contains a chimeric spectinomycin resistance gene (aadA) as selectable marker for isolation of transplastomic lines[65]. Regeneration of transplastomic lines on spectinomycin-containing medium resulted in the formation of plantlets that showed widely different pigmentation phenotypes. Green regenerants were obtained from some constructs, while the transplastomic lines obtained with other constructs displayed various degrees of pigment deficiencies, ranging from pale green to completely white (Fig. 1b; Fig. 2a, b).

The constructs expressing WAM1 and novispirin produced green plantlets. Leaves from these plantlets were subjected to another round of regeneration, and DNA was isolated from the resulting plants for restriction fragment length polymorphism (RFLP) analyses to confirm successful transformation of the chloroplast genome and assess the attainment of the homoplasmic state (i.e., the presence of a homogeneous population of transformed genomes and the absence of residual wild-type copies of the highly polyploid plastid genome[66]). Several lines were identified for which the RFLP analysis showed a strong band with the expected size suggesting correct integration of the transgene by homologous recombination (Supplementary Fig. 1).

Regenerated plantlets from transplastomic lines expressing 3-Molluxubin, 3-Novescandin and CXCL9 initially had variegated leaves with dark green, light green and/or white sectors, and RFLP analysis of this material revealed the simultaneous presence of hybridization signals for the transformed and the wild-type plastid genomes (heteroplasmy; Supplementary Fig. 1). After several additional regeneration cycles, individual regenerants showed apical formation of uniformly pale shoots and leaves, and RFLP analysis of DNA extracted from these phenotypically homogeneous tissues resulted in a single band corresponding in size to the transformed plastid genome (Supplementary Fig. 1). When the uniformly pale shoots were rooted, the pale phenotype remained stable over multiple propagation cycles, indicating that the lines had been purified to homoplasmy (Fig. 2a, b). Transplastomic plants expressing 3-Wamdeposin-5, 3-Wamdeposin-15, 6-MW-5, 6-MW-15, 9-MNW-5 and 9-MNW-15 resulted in completely white plants that ultimately also could reach homoplasmy (Supplementary Fig. 1; Fig. 2a).

As expected, white transplastomic plants were incapable of photoautotrophic growth when transferred from synthetic sugar-containing medium to soil and thus died quickly. This was also the case with several transplastomic lines that were very pale green such as the plants expressing the large single AMP CXCL9 (Fig. 2b). By contrast, plants expressing the single small HA-tagged AMPs WAM1 or novispirin were green and displayed only a slight delay in growth compared to wild-type plants (Fig. 2b).

Transplastomic plants with transgenes encoding fused AMPs (fAMPs) showed strong phenotypes, ranging from light green to completely white, and those that survived in soil exhibited extreme delays in growth and development (Fig. 2a). In some cases, the linker length influenced the plant phenotype, albeit not in a consistent manner. For 3-Molluxubin, the use of 15 amino acid linkers resulted in paler plants compared to those obtained by using the 5 amino acid linkers. By contrast, for 3-Novescandin, the 15 amino acid linkers permitted growth on soil, whereas the 5 amino acid linkers did not (Fig. 2a). 3-Novescandin-15 plants grew very slowly, producing seeds five to nine months after their transfer from in vitro culture to soil. Plants expressing 3-Molluxubin-5, 3-Molluxubin-15 and 3-Novescandin-5 could grow on soil only when grafted onto a wild-type stock that provided nutrients to the mutant shoots, a strategy that ultimately permitted seed production. Transplastomic plants expressing 3-Wamdeposin or any of the larger fusions (6-MW and 9-MNW) were white and unable to grow on soil, irrespective of the linker length (Fig. 2a), thus precluding the production of seeds.

The production of seeds from several lines enabled the additional assessment of the homoplasmic state of the transplastomic plants, by conducting inheritance assays[67]. When the transplastomic seeds were sown on spectinomycin-containing medium (Fig. 1b), no phenotypic segregation was observed, suggesting that parent plants and T1 offspring were homoplasmic.

## Expression of AMP and fAMP transgenes

Transgene expression at the mRNA level was assessed by northern blot analysis. All transgenes were expressed and accumulated mRNAs matching the expected sizes (Fig. 2c), in addition to some larger transcript species that likely originated from read-through transcription (a phenomenon that is pervasive in plastids and has been observed with transgenes inserted into the same genomic region in previous studies[68,69]).

Protein accumulation of AMPs and fAMPs (hereafter collectively referred to as (f)AMPs) was examined by immunoblotting using an antibody against the HA-tag. While no anti-HA signal could be observed for constructs expressing the single AMP novispirin as well as untagged CXCL9 or the large constructs combining 6 or 9 different AMPs, (Fig. 2d), all other transplastomic lines showed bands corresponding to the expected size of the respective AMPs and fAMPs, and additional weaker signals corresponding in size to possible multimers (Fig. 2d). A notable exception were WAM1 transplastomic plants that showed a strong signal at twice the expected size, possibly suggesting an unusually strong tendency to dimerize even under the denaturing conditions of purification and SDS-PAGE electrophoresis.

## Control of fAMP expression using the RNA amplification-enhanced riboswitch system

The observed phenotypes of plants constitutively expressing fused or large single AMPs suggested strong deleterious effects of AMP accumulation on plant growth and chloroplast development. To overcome this problem, inducible expression was attempted to allow for proper growth and assembly of the photosynthetic apparatus prior to fAMP accumulation. The chosen inducible RNA amplification-enhanced riboswitch (RAmpER) system[17] is entirely chloroplast based and controls transgene expression by a theophylline-inducible riboswitch[70]. Upon induction (i.e., watering of plants with theophylline solution), translational activation leads to the synthesis of the bacteriophage T7 RNA polymerase, which subsequently transcribes the gene of interest that is controlled by the T7 promoter[17] (PT7).

To eliminate the need to assemble large transformation vectors containing all elements of the RAmpER system (including the difficult-to-clone T7 RNA polymerase gene), we first generated a transplastomic line that harbors the riboswitch-controlled T7 RNA polymerase (RT7RP) in the chloroplast genome (Fig. 3b). This line (Nt-DK320) was used as recipient line for subsequent supertransformation experiments with constructs encoding the fAMP constructs under control of

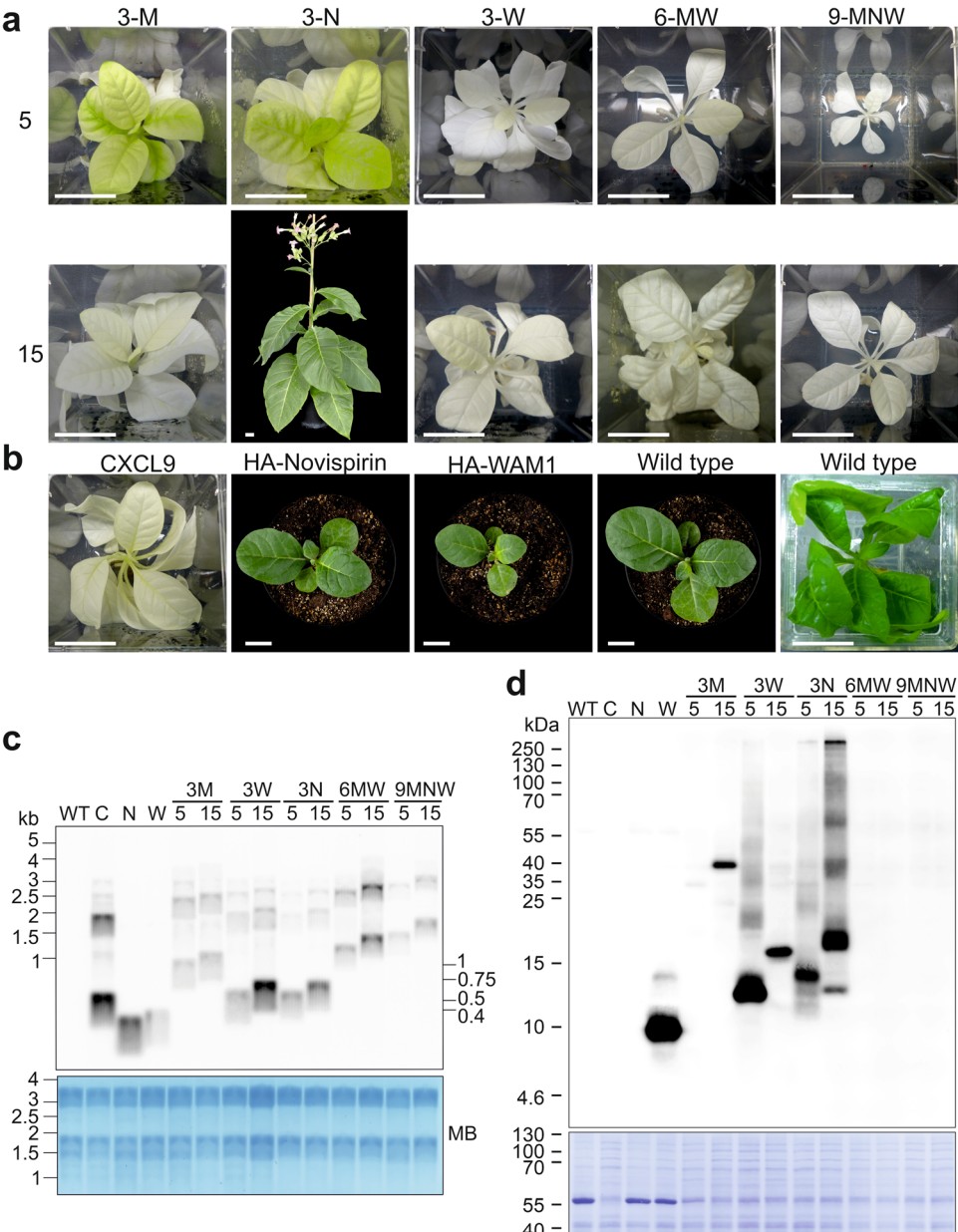

**Fig. 2 | Phenotypes of transplastomic plants that constitutively express AMPs and fAMPs and analysis of transgene expression. a** Plant phenotypes of transplastomic lines expressing fAMP combinations of 3, 6 or 9 different AMPs connected by 5 or 15 amino acid linkers (length of linker is indicated on the left of the pictures; cf. Supplementary Table 1, Supplementary Data 1 and 2). Plants incapable of photoautotrophic growth on soil are shown in tissue culture boxes. Photos were taken three months, or 4.5 months (6-MW-15) or six months (3-W-5) after transfer of (apical) shoot cuttings to fresh medium. 3-N-5 was the only fAMP line capable of growing on soil. The photo was taken five months after transfer from tissue culture. **b** Phenotypes of transplastomic plants expressing single AMPs. Shown are (from left to right) a CXCL9 plant three months after transfer of a shoot into a new box with synthetic medium, HA-WAM1, HA-Novispirin and a wild-type plant one month after sowing on soil, and a wild-type plant after one month of growth on synthetic medium. Scale bars: 2.5 cm. For images of HA-Novispirin, HA-WAM1 and wild-type

plants two months after sowing, see Supplementary Fig. 4. **c** Northern blot analysis to compare mRNA accumulation for AMP and fAMP transgenes in transplastomic tobacco lines (see Methods section). MB: methylene blue staining as loading control. The blot was performed twice with similar results. **d** Western blot analysis of AMP accumulation (see Methods section). Samples of total protein extracts (20 μg) were loaded, and fAMPs were immunodetected with an anti-HA antibody. Note that CXCL9 is not tagged. The Coomassie-stained membrane is shown below the immunoblot as a loading control. Three independent blots were performed with similar results. Plant material for northern and western blots was produced in sterile culture to enable side-by-side comparison to lines that do not survive on soil. C: CXCL9, N: HA-Novispirin, W: HA-WAM1, 3M: 3-Molluxubin, 3W: 3-Wamdeposin, 3N: 3-Novescandin, 6MW: 3M + 3W, 9MNW: 3M + 3N + 3W. Source data are provided as a Source Data file.

the *PT7* (Fig. 3a, b). The resulting supertransformed plants were named RAmpER::fAMP lines.

Some leakiness (i.e., basal expression in the non-induced state) has previously been reported for the RAmpER system[17,71]. To determine if this leakiness is caused by T7 RNA polymerase accumulation in the non-induced state or rather by transgene transcription from *PT7* even

in the absence of the T7 RNA polymerase, two control lines were generated. To this end, two of the constructs used for supertransformation of the RT7RP recipient line, were also used to transform wild-type plants. The resulting transplastomic lines were named PT7::fAMP, as opposed to the RAmpER::fAMP lines containing the RT7RP (Fig. 3).

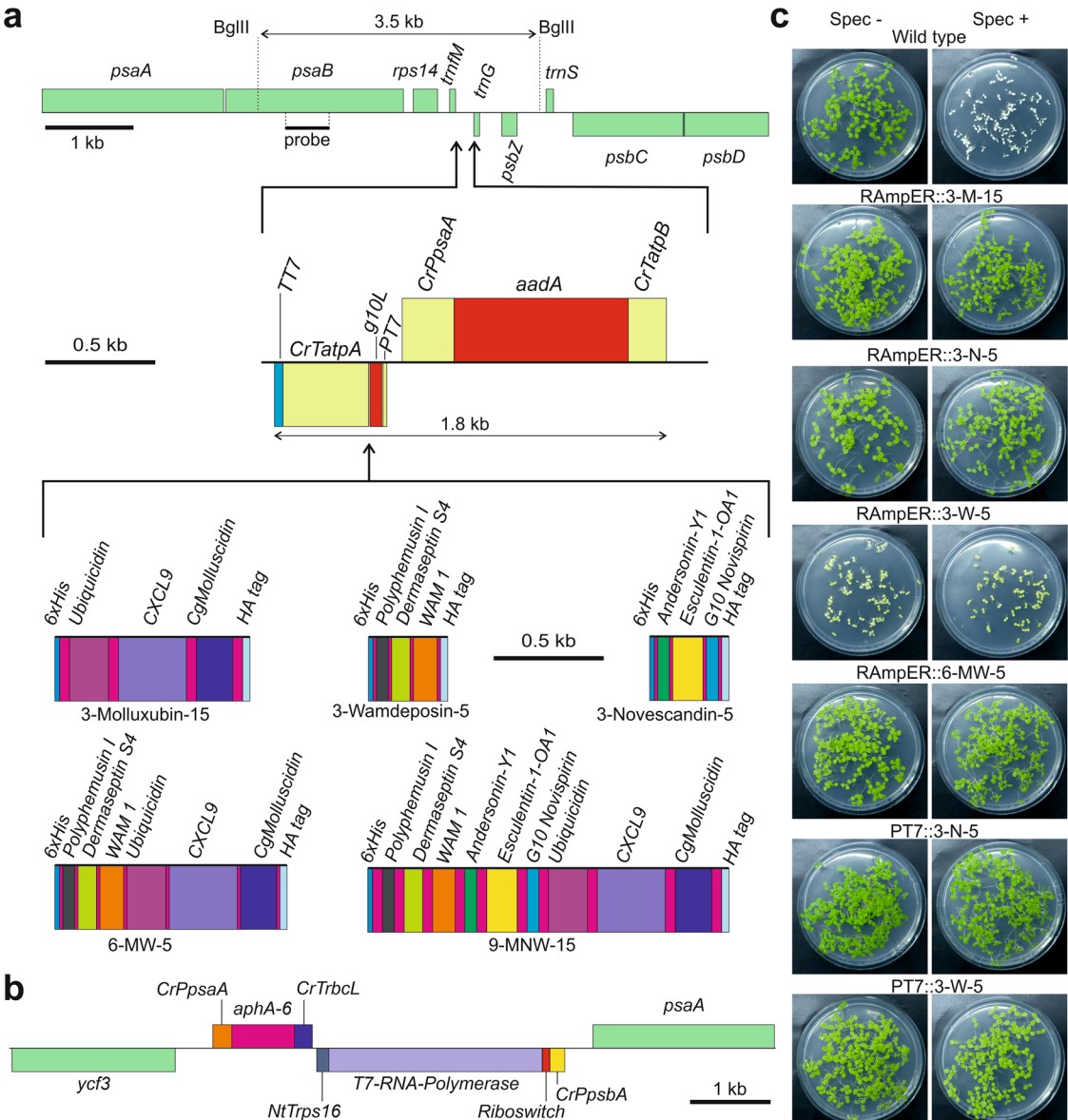

**Fig. 3 | Application of the RAmpER system to control expression of fAMPs from the tobacco plastid genome. a** Construct design (cf. Fig. 1a). The fAMP-encoding sequences were inserted between the T7 promoter (*PT7*) with the T7 *gene10* leader (*g10L*) and the *C. reinhardtii atpA* 3′ UTR (*CrTatpA*) followed by the T7 *gene10* 3′ UTR (*TT7*). For information on the fAMP combinations and the linkers (of 5 or 15 amino acids; pink), see Supplementary Table 1 and Supplementary Data 2. Note that the constructs driven by *PT7* have the opposite orientation in the plastid genome compared to the constitutive constructs (Fig. 1), to reduce potential read-through transcription from upstream plastid genes. **b** Physical map of the region in the plastid genome of the transplastomic recipient line Nt-DK320 that harbors the riboswitch-controlled T7 RNA polymerase gene targeted to the intergenic spacer between *ycf3* and *psaA* by selection for kanamycin resistance provided by a chimeric *aphA-6* gene controlled by the *C. reinhardtii psaA* promoter (*CrPpsaA*) and

*rbcL* 3′ UTR (*CrTrbcL*). The T7 RNA polymerase transgene is under the control of the *C. reinhardtii psbA* promoter (*CrPpsbA*), the theophylline-responsive riboswitch and the 3′ UTR from the tobacco *rps16* gene (NtTrps16). Supertransformation of transplastomic plants harboring the T7 RNA polymerase cassette with *PT7*-driven fAMP transgenes was performed to create transplastomic lines with RAmpER-controlled fAMP expression. **c** Seed tests of transplastomic lines expressing fAMPs under RAmpER or *PT7* control. Surface sterilized seeds were sown on medium with or without spectinomycin (Spec). Photos were taken after two weeks, the wild-type controls are also used in Fig. 1b. RAmpER::3-W-5 plants were incapable of autotrophic growth on soil, and therefore, seeds were obtained by grafting a shoot raised in sterile culture onto a wild-type stock. 3-M-15: 3-Molluxubin-15, 3-N-5: 3-Novescandin-5, 3-W-5: 3-Wamdeposin-5, 6-MW-5: 3-M-5 + 3-W-5.

---

The homoplasmic state for the transplastomic RAmpER::fAMP and PT7::fAMP plants (Fig. 3a) was confirmed by RFLP analysis (Supplementary Fig. 2) and seed test (Fig. 3c) as described before.

### Phenotypes of RAmpER::fAMP plants
Transplastomic plants in which production of fAMPs was controlled by the RAmpER system showed markedly improved growth phenotypes in the non-induced state compared to strong constitutive fAMP expression, in all cases. However, they still did not display

wild-type-like pigmentation and growth rate (Fig. 4a). Replacement of the constitutive expression signals by RAmpER effectively shifted the plant phenotypes from pale green to darker green or from white to pale green (Fig. 4a). RAmpER::3-Molluxubin-15 and RAmpER::3-Novescandin-5 were green and capable of growing autotrophically on soil, albeit at slower rate than wild-type plants. While Prrn::3-Wamdeposin-5 and Prrn::9-MNW-15 plants were white, RAmpER::3-Wamdeposin-5 and RAmpER::9-MNW-15 plants were pale green, but still unable to grow on soil. A graft of RAmpER::3-Wamdeposin-5 onto a

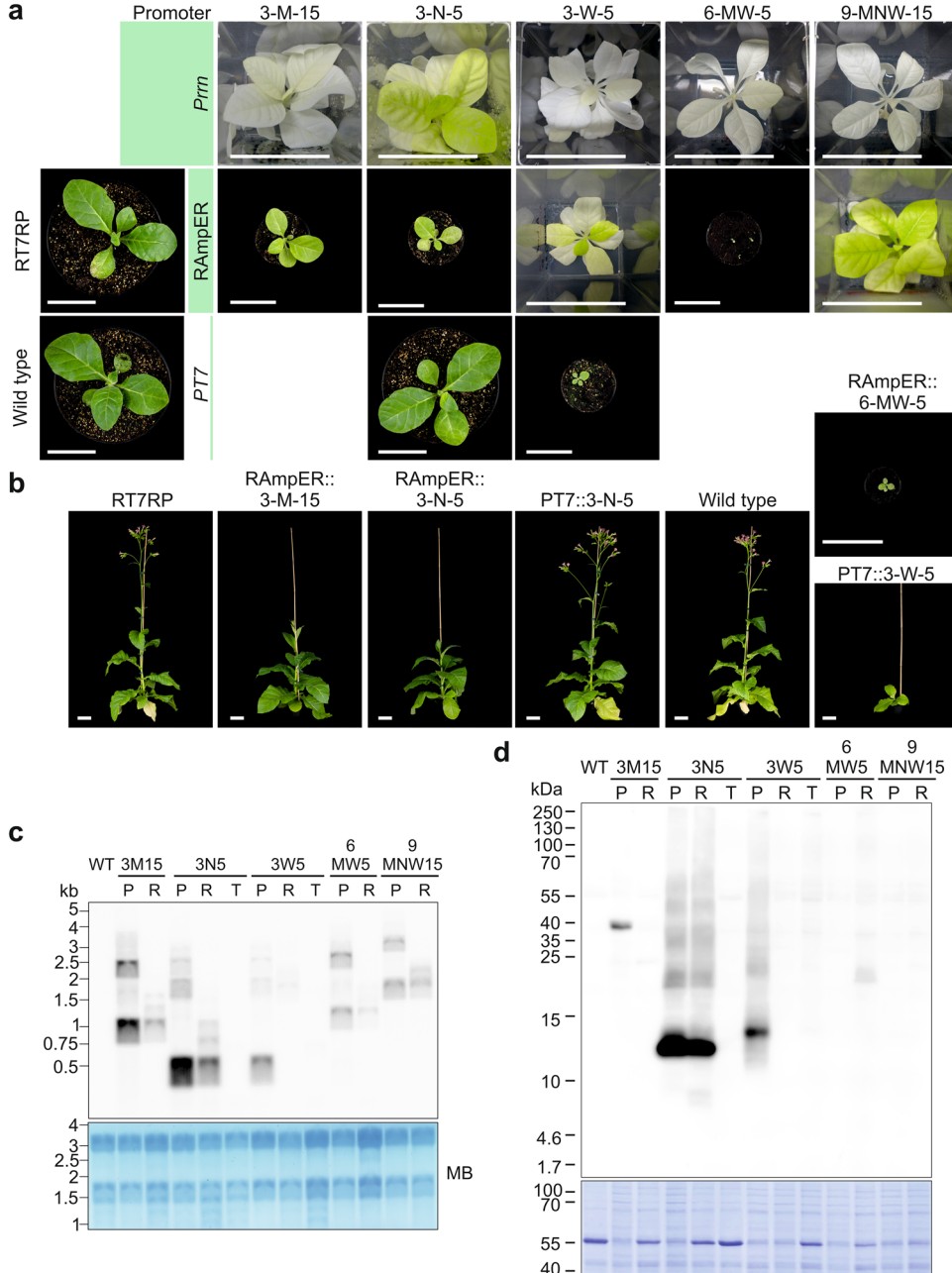

**Fig. 4 | Comparison of transplastomic plants expressing fAMP constructs controlled by different promoter systems. a** Phenotypes of transplastomic plants. Two genotypes were used as recipients for introduction of fAMP constructs: the wild type and a transplastomic line harboring the riboswitch-controlled T7 RNA polymerase (RT7RP). The promoters used to drive fAMP expression is given in each row. Shown are transplastomic plants expressing fAMPs under the control of the Prrn promoter (images from Fig. 2a), transplastomic plants expressing fAMPs under RAmpER control (uninduced), and two PT7-controlled fAMP lines in the wild-type background. Photos of soil-grown plants were taken one month after sowing. Scale bars: 5 cm. 3-M: 3-Molluxubin, 3-N: 3-Novescandin, 3-W: 3-Wamdeposin, 6-MW: 3-M + 3-W, 9-MNW: 3-M + 3-N + 3 W. The numbers 5 or 15 indicate the lengths of the amino acid linkers separating the AMPs (see Supplementary Table 1 and Supplementary Data 2). **b** Images of the plants in (a) two months after sowing. Scale bars: 5 cm. **c** Northern blot analysis comparing the fAMP expression levels in Prrn

(P), RAmpER (uninduced) (R) and PT7 (T) plants (see Methods). MB: methylene blue staining (loading control). This blot was performed twice with similar results. **d** Western blot analysis of fAMP accumulation. Strong constitutive expression from Prrn (P) is compared with basal RAmpER expression (in the non-induced state; R) and low-level background expression from PT7 (in the absence of the T7 RNA polymerase; T). Total protein samples (20 μg) were loaded, and fAMPs were immunodetected with an anti-HA antibody. The Coomassie-stained membrane is shown below the immunoblot as loading control. Note that pale and white lines have reduced amounts of the large subunit of RuBisCO (at ~55 kDa). Two independent blots were performed with similar results. Plant material for northern and western blots was produced in sterile culture to enable inclusion of lines that do not survive on soil. 3M15: 3-Molluxubin-15, 3N5: 3-Novescandin-5, 3W5: 3-Wamdeposin-5, 6MW5: 3M5 + 3W5, 9MNW15: 3M15 + 3N15 + 3W15. Source data are provided as a Source Data file.

wild-type stock survived on soil, produced seeds, and could therefore be included in the seed assays (Fig. 3c). While Prrn::6-MW-5 plants were white, RAmpER::6-MW-5 plants were green, grew on soil and produced seeds, although growth was severely retarded (Fig. 4).

When grown side by side, from seeds, PT7::3-Novescandin-5 plants were indistinguishable from wild-type plants, while RAmpER::3-Novescandin-5 plants showed a delayed growth. PT7::3-Wamdeposin-5 plants grew on soil and were green, unlike the RAmpER::3-

Wamdeposin-5 lines (Fig. 4). In contrast to PT7::3-Novescandin-5, PT7::3-Wamdeposin-5 showed a clear delay in growth (Fig. 4).

## fAMP accumulation in RAmpER and PT7 plants

Northern and western blot analyses were performed to compare the levels of fAMP mRNA and protein accumulation in non-induced RAmpER::fAMP and PT7::fAMP plants with those in the corresponding constitutively expressing Prrn::fAMP plants. As expected and in agreement with the phenotypes of the lines (Fig. 4a), fAMP mRNA and protein levels were the highest in Prrn::fAMP plants, substantially lower in non-induced RAmpER::fAMP plants and undetectable in PT7::fAMP plants (Fig. 4c, d).

Initial extensive attempts to purify fAMPs from RAmpER::fAMP plants were unsuccessful. We found that solubilization of the fAMPs from plant tissues required the use of detergents, and subsequent separation of proteins and detergent was unsuccessful. The presence of detergents in the protein extracts is highly undesirable, because they themselves have antimicrobial activity and thus would interfere with any downstream test for biological activity.

## SUMO fusions for enhanced solubility and reduced toxicity of fAMPs

To improve fAMP solubility while potentially reducing their toxicity towards the host, fusions of the fAMPs to the small ubiquitin-like modifier (SUMO) were constructed. To this end, vectors were produced for C-terminal fusion of the protein of interest to N-terminally 6xHis-tagged SUMO (12.8 kDa, 111 amino acids) under the control of the *T7* or *Prrn* promoter (Fig. 5a, b).

Seven SUMO fusion constructs with different expression elements were generated (Fig. 5a, b). Constructs with *PT7* were introduced in both the RT7RP recipient line to generate RAmpER::SUMO-(f)AMP plants and wild-type plants. Transplastomic plants were obtained, and their homoplasmic state was confirmed by RFLP analysis and seed test as described previously (Supplementary Fig. 3 and Fig. 5c).

## Phenotypes of transplastomic plants expressing SUMO-(f)AMPs

Fusing 3-Wamdeposin-5 to SUMO, in conjunction with reduced expression levels, reduced the toxicity of this polypeptide to the plant. RAmpER::SUMO-3-Wamdeposin-5 plants were visibly darker green than plants expressing the same construct without the SUMO tag (Fig. 6a). Similarly, PT7::SUMO-3-Wamdeposin-5 plants were considerably less retarded in growth than the PT7::3-Wamdeposin-5 plants (Fig. 6a). By contrast, Prrn::SUMO-3-Wamdeposin-5 plants were completely white, resembling the phenotype of Prrn::3-Wamdeposin-5 plants.

Although RAmpER::SUMO-3-Wamdeposin-5 plants were unable to grow autotrophically, seeds could be obtained from a heteroplasmic plant that developed a shoot exhibiting the pale color typical of this construct upon segregation into homoplasmy.

For single AMPs, contrasting effects of SUMO fusion were observed. Whereas growth of Prrn::HA-WAM1 plants was only mildly retarded, Prrn::SUMO-HA-WAM1 plants grew slowly and were pale green (Fig. 6). However, when the latter construct was controlled by RAmpER, this phenotype was much alleviated and only consisted of a slight growth delay (Fig. 6a). RAmpER::SUMO-HA-WAM1 plants in the T0 and T1 generations were male sterile (4 independently generated transplastomic lines) and had to be pollinated with wild-type pollen to obtain seeds.

While Prrn::HA-Novispirin plants were nearly indistinguishable from wild-type plants, Prrn::SUMO-HA-Novispirin plants grew slower and new leaves were pale green (Fig. 6a), presumably because SUMO fusion caused increased AMP accumulation. RAmpER::SUMO-GFP plants displayed wild-type-like growth, while Prrn::SUMO-GFP plants were delayed in growth, presumably due to other mechanisms than in

the SUMO-AMP plants, like exhaustion of the protein synthesis capacity[22].

## Accumulation of SUMO fusion proteins

The improved growth of RAmpER::SUMO-3-Wamdeposin-5 compared to RAmpER::3-Wamdeposin-5 plants could be due to reduced fAMP accumulation upon fusion to SUMO. However, immunoblot analysis revealed the opposite, as a much stronger signal was seen for RAmpER::SUMO-3-Wamdeposin-5 than for RAmpER::3-Wamdeposin-5 (Fig. 6c). For all fAMPs tested, the SUMO fusion proteins accumulated to higher levels than the unfused versions (Fig. 6c).

No consistent difference was observed in the steady-state abundance of the transgenic transcripts between the versions with and without SUMO, suggesting that the observed increase in protein accumulation is caused by improved stability of SUMO fusion proteins (Fig. 6b, c).

## Purification of SUMO fusion proteins

Protein purification was performed successfully for SUMO-HA-WAM1, SUMO-HA-Novispirin and SUMO-GFP using buffers without detergents. All three proteins were detected in the soluble phase, indicating that fusion to SUMO improves solubility of the AMPs, thus obviating the need to use detergents (Fig. 7a). In Prrn::SUMO-GFP and RAmpER::SUMO-GFP plants, SUMO-GFP was readily detectable by Coomassie staining (Fig. 6c) and accumulated to similar levels as the most abundant endogenous protein, the large subunit of RuBisCO.

Batch purification with Ni-NTA agarose yielded full-length SUMO-GFP in the eluate (Fig. 7a). This was also the case with SUMO-HA-WAM1 and SUMO-HA-Novispirin, although an additional band corresponding in size to free SUMO was present in the eluate from the SUMO-HA-WAM1 purification experiments (Fig. 7a). Large-scale purification followed by quantification of the purified protein products revealed yields of approximately 56 μg per g leaf material for Prrn::SUMO-HA-WAM1, 54 μg per g leaf material for Prrn::SUMO-HA-Novispirin, 41 μg per g leaf material for RAmpER::SUMO-HA-WAM1 (non-induced), and 654 μg per g leaf material for Prrn::SUMO-GFP. Based on ratios between the molecular weights of 6xHis::SUMO (12,794 Da) and the AMPs or protein fused to it (HA-WAM1: 5,864 Da, HA-Novispirin: 3,794 Da, GFP: 27,056 Da), we also calculated the yields without the SUMO part of the recombinant proteins. For Prrn::SUMO-HA-WAM1, yields were approximately 18 μg of HA-WAM1 per gram leaf material, for RAmpER::SUMO-HA-WAM1 (non-induced), yields were approximately 13 μg HA-WAM1 per gram leaf material, for Prrn::SUMO-HA-Novispirin, yields were approximately 12 μg HA-Novispirin per gram leaf material, and for Prrn::SUMO-GFP, yields were approximately 444 μg GFP per gram leaf material. Quantifications are provided in Supplementary Data 3.

Next, we tested whether the proteins of interest could be released by cleavage of the purified fusion proteins with recombinant SUMO protease, a highly specific enzyme which hydrolyzes the peptide bond at the C-terminus of the SUMO domain. After incubation with the protease, the expected size shift was observed that corresponded to conversion of full-length SUMO-tagged protein to free SUMO (Fig. 7b). The efficiency of digestion based on image quantification is approximately 80% for SUMO-GFP and SUMO-Novispirin, and 60% for SUMO-HA-WAM1. Quantifications are provided in Supplementary Data 3. For digested SUMO-GFP, free GFP was readily detected on the Coomassie-stained western blot membrane. No band was seen for HA-WAM1 by Coomassie staining, but the peptide was observed by immunodetection (Fig. 7b). After cleaving SUMO-HA-Novispirin a low-molecular weight band was detected on the Coomassie stained western blot membrane, which migrated slightly slower than the predicted size of HA-Novispirin. However, this band was recognized only weakly by the anti-HA antibody (Fig. 7b).

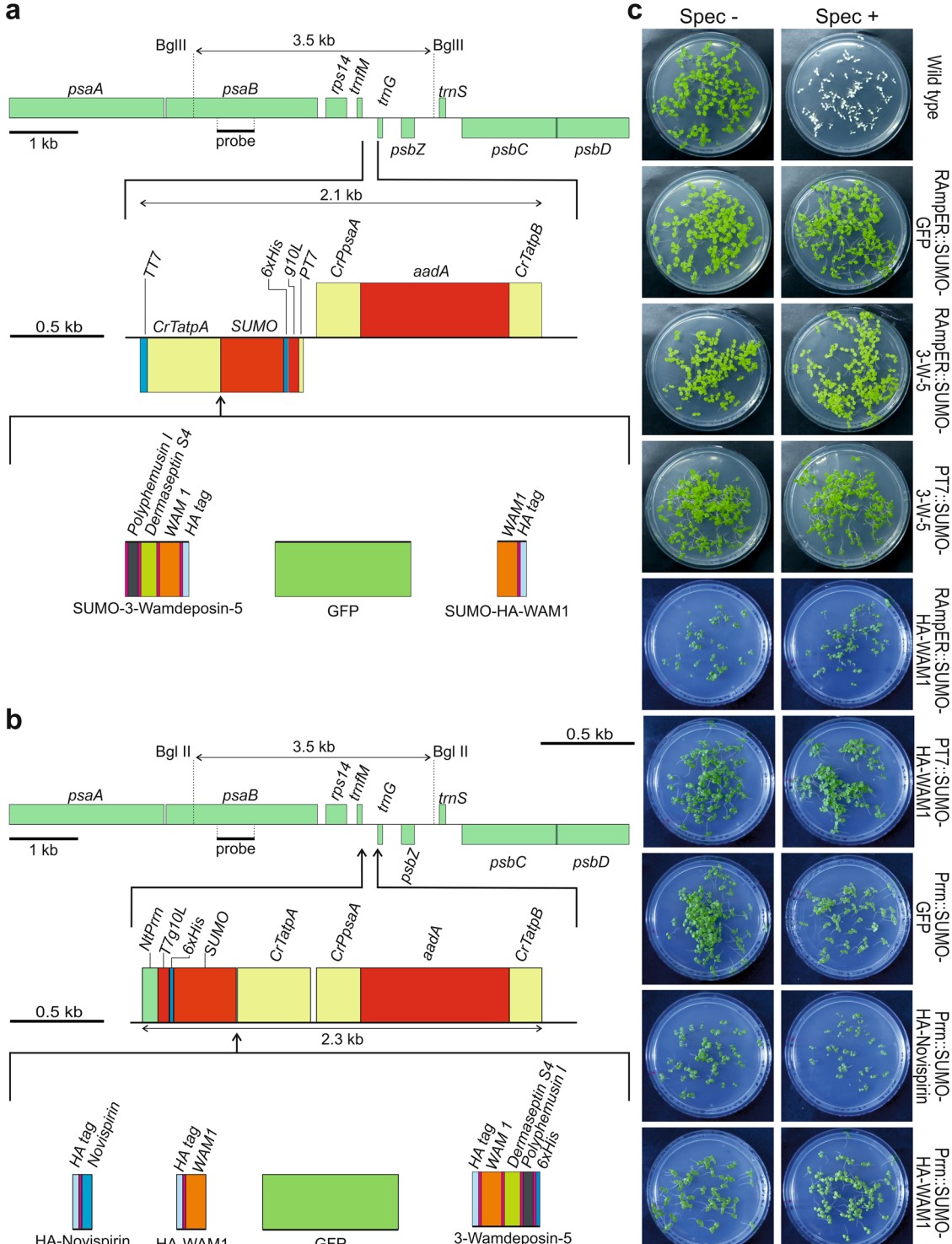

**Fig. 5 | Transplastomic expression of AMPs and fAMPs as SUMO fusions.**
**a** Physical maps of constructs for the expression of 6xHis-SUMO fusions under the control of the T7 RNA polymerase promoter (*PT7*). For information on construct design and the expression elements used, see Figs. 1a and 3a. **b** Physical maps of constructs for the expression of 6xHis-SUMO fusions under the control of the plastid rRNA operon promoter (Prrn) and the *gene10* leader (*T7g10L*) from phage T7. Note that the transgenes under the control of *PT7* have the opposite orientation in the plastid genome in comparison to the *Prrn*-driven transgenes, to reduce potential read-through from upstream plastid genes. **c** Seed tests to confirm homoplasmy of transplastomic plants expressing SUMO fusions. Because RAmpER::SUMO-3-Wamdeposin-5 plants were unable to survive on soil, seeds were obtained from a heteroplasmic plant that segregated into dark green (wild-type-like) and pale leaves and branches. Seeds were harvested from pods of the pale branches.

## Antibacterial activity of chloroplast-produced AMPs

Having established purification procedures for AMPs fused to SUMO, we next wanted to examine, if these proteins have antibacterial activity. To this end, the purified protein samples were tested in radial diffusion assays (Fig. 7c; see Methods). To determine if removal of SUMO was required for antimicrobial activity, both the intact fusion proteins and the protease-cleaved proteins were tested. As expected, SUMO-GFP (incubated with or without SUMO protease) did not inhibit

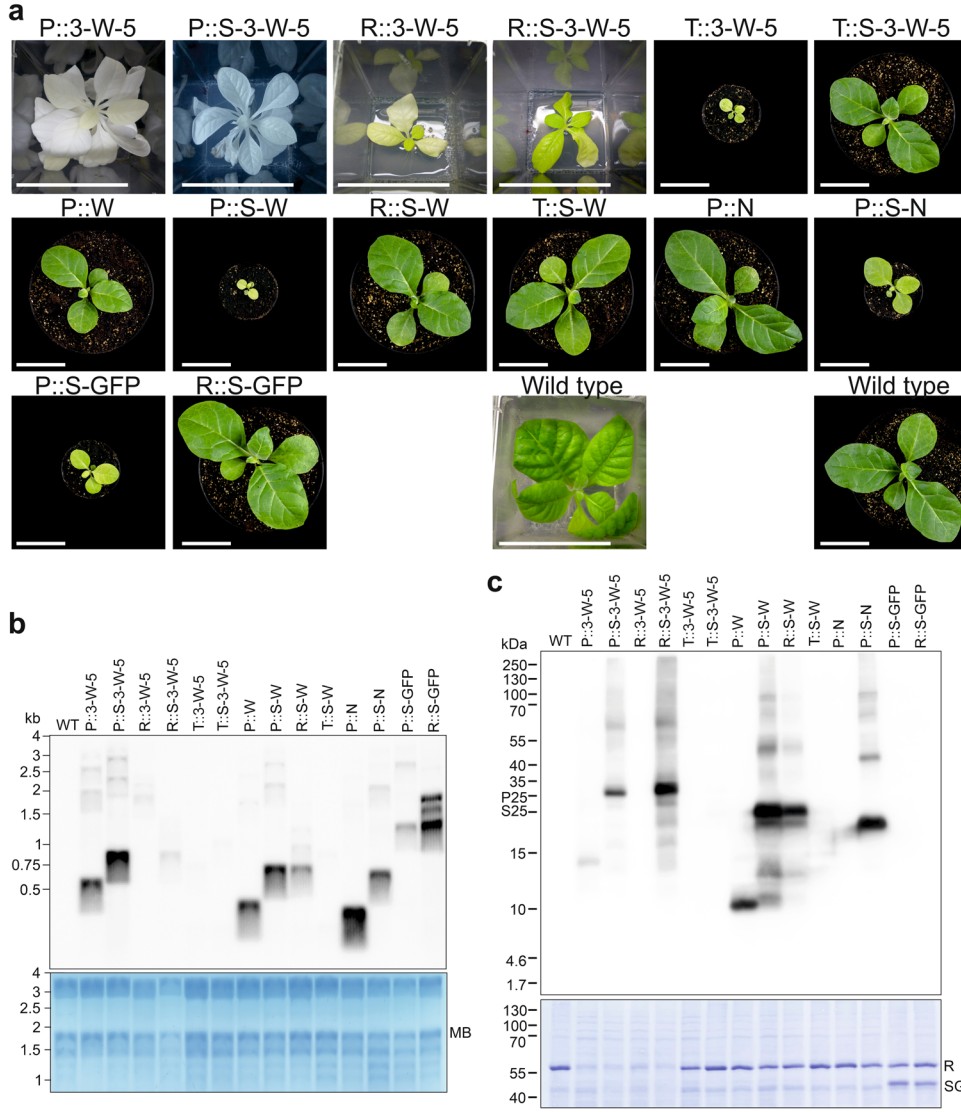

**Fig. 6 | Comparison of transplastomic plants expressing AMPs and fAMPs either alone or as fusions to SUMO (S) under the control of different promoter systems. a** Images of plants grown in sterile culture on synthetic medium. Photos were taken three weeks after transfer of stem cuttings from R::3-W-5 and R::S-3-W-5, six months after transfer for P::3-W-5 (image from Fig. 2a), and three months after transfer for P::S-3-W-5. Photos of soil-grown plants were taken one month after sowing. Scale bars: 5 cm. For images of soil-grown plants after two months, see Supplementary Fig. 4. **b** Northern blot analysis comparing mRNA accumulation levels in the different transplastomic lines shown in (a) (see Methods section). MB: methylene blue staining (as loading control). This blot was performed twice with similar results. **c** Western blot analysis comparing the effects of SUMO fusion on protein accumulation in the different transplastomic lines. Samples of 20 μg total leaf protein were separated by gel electrophoresis, blotted and immunodetection

was performed with anti-HA antibody. The Coomassie-stained membrane is shown below the immunoblot as a loading control. This blot was performed once, but results were confirmed for RAmpER::3-W-5 by an independent blot. Note that SUMO-GFP (SG) accumulates to similarly high levels as the large subunit of RuBisCO (R) in both Prrn::SUMO-GFP (P::S-GFP) and RAmpER::SUMO-GFP (R::S-GFP) plants. Note that the SUMO-GFP protein is not detected in the western blot, due to absence of an HA-tag. S25: 25 kDa band of the Spectra™ Multicolor Low Range Protein Ladder (1.7 to 40 kDa; ThermoFisher); P25: 25 kDa band of the PageRuler™ Plus Prestained Protein Ladder (10 to 250 kDa; ThermoFisher). RNA and protein was extracted from plants grown on synthetic medium to enable inclusion of lines that did not survive on soil. Source data are provided as a Source Data file.

bacteria (*Escherichia coli*) growth (Fig. 7c). By contrast, purified SUMO-HA-WAM1 and SUMO-Novispirin displayed clear antimicrobial activity (Fig. 7c). Interestingly, no difference in antimicrobial activity was observed between the samples treated with and without SUMO protease, suggesting that fusion to SUMO does not appreciably affect the activity of AMPs.

## Discussion

In the face of rapidly spreading resistance of pathogenic bacteria to classical antibiotics, the development of new types of antimicrobial agents is becoming one of the most pressing global health challenges and, in addition, is of great importance to plant production and animal

husbandry. In the course of this work, we have explored diverse strategies to utilize transplastomic tobacco plants as production platform for antimicrobial peptides (Supplementary Table 2).

Chloroplasts can express foreign proteins to very high levels[25], including large antibacterial proteins such as phage-derived lytic enzymes (so-called endolysins[22,72]) and antiviral agents[24,73]. However, the expression of very small polypeptides poses serious challenges, due to the presence in plastids of specific proteases that degrade polypeptides smaller than 65 amino acids[36,37]. We have addressed this problem by constructing protein fusions that increase AMPs size above the critical threshold for protease recognition. This was achieved by fusion of multiple AMPs separated by flexible linkers

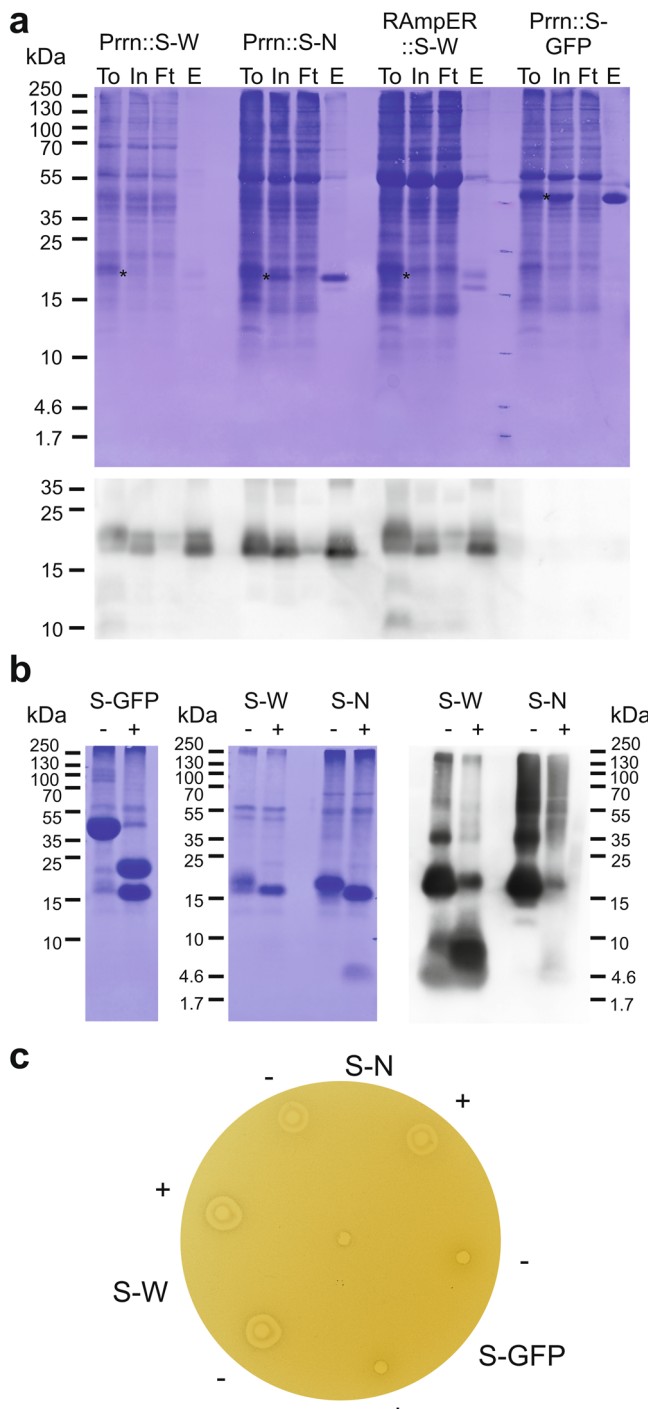

**Fig. 7 | Purification and antimicrobial activity of SUMO-AMP fusion proteins from transplastomic plants. a** Ni-NTA agarose purification of SUMO-AMP and SUMO-GFP fusion proteins based on the 6xHis tag. Protein samples were separated by gel electrophoresis and blotted. The western blot was stained with Coomassie prior to blocking and immunodetection with an anti-HA antibody. For extraction, frozen leaf material was ground and mixed with extraction buffer. An aliquot of this suspension was directly loaded as total protein (To; incubated with denaturing sample buffer for 5 min at 95 °C and insoluble material removed by centrifugation). The insoluble material was then removed by centrifugation, and the supernatant was used as input for purification (In) by incubation with Ni-NTA agarose beads. The unbound flow through fraction was collected (Ft), and after washing the column, the bound SUMO fusion proteins were eluted (E). Asterisks indicate the position of the fusion proteins in the To and In fractions. Two independent blots were performed with similar results. **b** Cleavage assay of the purified SUMO fusion proteins using SUMO protease. 170 μg SUMO-GFP (S-GFP), 45 μg SUMO-HA-WAM1 (S-W), and 96 μg SUMO-HA-Novispirin (S-N) were incubated with (+) or without (−) 25 units of SUMO protease (Ulp1: 27 kDa). Of these samples, 35.5 μg SUMO-GFP, 9 μg SUMO-HA-WAM1, and 20 μg SUMO-HA-Novispirin were separated by SDS-PAGE followed by western blotting. The resulting membranes were stained with Coomassie followed by immunodetection of S-W and S-N with an anti-HA antibody. Two independent blots were performed with similar results. **c** Activity test of purified AMPs by radial diffusion assays. Equimolar concentrations (150 μM) of SUMO fusions (in 20 mM Tris pH 8.0 buffer) were incubated with or without 2.5 units of SUMO protease, and applied in an antimicrobial activity radial diffusion assay. Antimicrobial activity is observed as a clear zone without bacterial growth around the well. The well in the center of the plate was loaded with the buffer control. Three independent assays were performed with similar results. For abbreviations, see (**b**). Source data are provided as a Source Data file.

expressed in bacteria[76]. The use of SUMO as carrier protein in chloroplasts is an approach that has not been explored before. Our data suggest that fusion to SUMO provides a valuable tool for efficient recombinant protein production and purification from transplastomic plants. In our study, SUMO fusion had three beneficial effects. First, in all (f)AMPs tested, we observed improved protein accumulation, presumably by increasing protein stability. Second, it greatly improved solubility of the AMPs, thus facilitating purification and bypassing the need to use detergents for AMP extraction. Third, combined with lowered expression levels, fusion to SUMO also alleviated the mutant phenotypes associated with fAMP expression.

Previous work has shown that recombinant proteins that integrate into or interact with membranes impair chloroplast development, thus causing pigment deficiency and growth defects[35]. As AMPs have a strong propensity to interact with membranes[1,3,77], the mutant phenotypes observed in many of our transplastomic (f)AMP-expressing plants were not entirely surprising (Figs. 2a, 4a and 6a). We have shown that the use of an inducible expression system and/or fusion to SUMO can alleviate these defects, although in most cases, it cannot restore wild-type-like growth. However, optimization of expression constructs for several (f)AMPs resulted in plants that display only moderate growth retardation compared to the wild type (Figs. 4b and 6a). Given that AMPs are high-value products, such small delays in growth are likely acceptable for commercial production. In the non-induced state, the RAmpER system behaved as a moderately strong promoter resulting in (f)AMP accumulation with reduced negative impact on plant growth. For increased production, induction with theophylline at the optimized time and concentration could further increase the production[17,71].

An additional product from the research reported here was the generation of a transplastomic line harboring the riboswitch-controlled T7 RNA polymerase gene (RT7RP line, Fig. 3b). Since this line does not contain *aadA*, the standard marker gene for chloroplast transformation, it represents an ideal recipient line for super-transformation with transgenes for inducible expression using the *aadA* marker and spectinomycin selection (as demonstrated with our series of RAmpER::fAMP transplastomic lines; Figs. 3 and 5; Supplementary Figs. 2 and 3; Supplementary Table 2 and Supplementary

and/or fusion to SUMO, a soluble protein that can be easily removed post-synthesis by cleavage with a highly specific protease. In our work, fusion to the full-length SUMO protein enabled efficient AMP release by proteolytic cleavage. Although the majority of SUMO fusion was efficiently cleaved, part of the purified product may be in a conformation that did not enable recognition by the SUMO protease. SUMO cleavage mimics SUMO maturation in yeast, where the SUMO protease recognizes the structure of SUMO and cleaves within the C-terminal sequence motif GG/ATY converting full-length pre-SUMO into the mature form[74–76]. To obtain a final product without any additional amino acids, the ATY sequence could also be omitted, in which case cleavage takes place between the C-terminal GG of SUMO and the fused protein sequence downstream (except if the GG sequence is followed by a proline residue), as demonstrated for SUMO fusions

Data 2). This RT7RP recipient line will greatly simplify construct assembly in future studies using the RAmpER system for transgene expression in plastids.

In our work, yields of up to 56 µg SUMO-AMP per g leaf material were obtained, corresponding to 18 µg of HA-WAM1 and 12 µg of HA-Novispirin. Previous studies on the production of cationic AMPs from the plastid genome reported production of AMP fusions to GFP with yields of 5 µg per g fresh leafy biomass (chromatography purification) or 53 µg per g (organic extraction) for a Retrocyclin-GFP fusion and 8 µg per g for a Protegrin-1-GFP fusion[26]. Production of cationic AMPs in plants by stable genetic transformation of the nuclear genome yielded up to 7 µg per g fresh weight (for MsrA2[27,28]). Production of AMPs on the surface of a recombinant plant virus that was used to infect plants yielded up to 25 µg/g fresh weight of AMP (for SP1-1[27,29]). By contrast, anionic AMPs fused to the elastin protein and expressed transiently in plants appear to accumulate to higher levels than cationic AMPs, reaching yields of up to 563 µg/g FW, corresponding to 113 µg/g FW of AMP (for ADP2[27,32]). However, cationic AMPs are more versatile antimicrobial agents than anionic AMPs, because they have a stronger binding affinity to microbial membranes, due to the abundant presence of anionic compounds in the membranes of bacteria and fungi (including lipopolysaccharides in gram-negative bacteria, lipoteichoic acid in gram-positive bacteria and mannans in fungi[78]).

In our work, we demonstrated activity of plastid-produced HA-WAM1 and HA-Novispirin against the gram-negative bacterium *E. coli*. Previous studies had demonstrated that WAM1 and Novispirin are active against both gram-negative and gram-positive bacteria[56,79] (Supplementary Data 1), and given the low folding requirements of AMPs, their broad activity is expected to be the preserved in the plastid-produced peptides. A particularly interesting finding from our antibacterial activity assays was that the bactericidal activity of AMPs was independent of SUMO removal. This observation suggests that folding of AMPs into their active three-dimensional conformation is not affected by fusion to a comparatively large N-terminal protein domain. It also offers the possibility to use AMPs as fusions to other functional proteins in practical applications, thus avoiding the need for protease cleavage and providing the option to work with proteins that are more manageable in size, multifunctional and potentially more stable.

AMPs are also naturally produced by plants and have been proposed to constitute an important part of plant immunity[80–82]. Future work will be directed towards the construction of tighter inducible expression systems that allow the generation of AMP-synthesizing plants with no negative impact on plant growth. In addition to providing a potential source of AMPs for pharmaceutical purposes, such plants could be tested for resistance to plant-specific bacterial and fungal pathogens, potentially providing a powerful tool for crop protection against diseases in agriculture.

## Methods

### Design of expression constructs for fusions of antimicrobial peptides

The selected AMP sequences were linked to each other with either 5 or 15 amino acid linkers composed of glycine and serine. An N-terminal human influenza hemagglutinin (HA) tag (MGYPYDVPDYA) and a C-terminal 6xHis-tag were added, also connected by 5 or 15 amino acid linkers. DNA sequences encoding these fused AMP amino acid sequences were generated by digital reverse translation and optimized to the preferred codon usage of tobacco chloroplasts[63]. The DNA sequences encoding the linkers were adjusted to the codon usage in tobacco chloroplasts[63], in such a way that any repetition of 10 or more nucleotides between the different linkers was avoided (Supplementary Table 1 and Supplementary Data 2), to reduce the risk of unwanted recombination within the chloroplast genomes[61,83]. For the 5 amino acid linkers, this was possible with the amino acid sequence GGSGG,

however, for the 15 amino acid linkers, the amino acid sequence had to be varied (Supplementary Table 1 and Supplementary Data 2). For N-terminal HA-tagging, a single glycine was used as mini linker between the initiator methionine and the HA-tag. Glycine was selected as penultimate amino acid based on previous success with the expression of griffithsin[24]. Nucleotide sequences of 15 nt identical to the flanking regions of the insertion site in the chloroplast transformation vectors were included in the synthesized DNA to facilitate homology-based cloning.

### Generation of plastid transformation constructs

To achieve high-level constitutive expression of AMP constructs, plasmid pDK323 (Fig. 1a) was used to place the genes of interest under the control of the tobacco plastid ribosomal RNA operon promoter (*Prrn*) fused to the T7 phage *gene10* leader (g10L) and the 3′ UTR of the chloroplast *atpA* gene from *Chlamydomonas reinhardtii* (*CrTatpA*). pDK323 also contains the spectinomycin resistance gene *aadA* that encodes the enzyme aminoglycoside 3′′-adenylyltransferase[65] and is driven by the *C. reinhardtii psaA* promoter. In addition, the vector contains flanking homology regions for integration into the plastid genome between the *trnfM* and *trnG* genes[64] by homologous recombination (Fig. 1).

For inducible expression from the plastid genome, plasmid pMPH36 was created. pMPH36 is similar to pDK323 that was used for constitutive expression, but contains the T7 promoter instead of the *Prrn* promoter and harbors the T7 promoter-driven transgene in opposite orientation to *trnG* (see Figs. 1, 3 and 5 for construct maps). pMPH36 also contains flanking regions for integration between *trnfM* and *trnG* in the plastid genome by homologous recombination. Constructs derived from pMPH36 were used to supertransform pDK320 (Fig. 3b) transplastomic recipient plants that carried a riboswitch-controlled T7 RNA polymerase gene[17] (RT7RP) in the plastid genome between the genes *ycf3* and *psaA*. RT7RP transplastomic plants were generated by using the *aphA6* gene[84] (encoding aminoglycoside phosphotransferase) providing kanamycin resistance as selection marker (Fig. 3). This strategy allowed the use of *aadA* as marker for subsequent supertransformation of RT7RP plants with pMPH36-derived constructs, resulting in plants with a functional RNA amplification-enhanced riboswitch (RAmpER) system[17]. Some expression cassettes controlled by the T7 promoter were additionally used for transformation of wild-type plants.

DNA sequences for the different AMPs were synthesized (GeneCust) and used as template for PCR. PCR products were generated with Phusion High-Fidelity DNA Polymerase (Thermo Fisher Scientific), gel purified (Nucleospin® Gel and PCR Clean-up kit; Macherey-Nagel) and inserted into EcoRV linearized vectors (pDK323, pMPH36) using In-Fusion® HD Cloning (Takara) (Figs. 1, 3 and 5). To form larger fusions of six or nine AMPs, different constructs comprising three peptides each were joined by overlap-extension PCR followed by In-Fusion® HD Cloning.

To construct SUMO fusions, the DNA sequence encoding the full-length (except for the last amino acid) yeast SUMO (SMT3; NCBI Reference Sequence: NP_010798.1) with an N-terminal 6xHis tag (MEHHHHHHGG-) was codon optimized according to the preferred codon usage in chloroplasts[63] and synthesized (Invitrogen). The sequence encoding SUMO ended with a SnaBI restriction site which was used for plasmid linearization. A glutamate residue was included between the initiator methionine and the 6xHis-tag (Supplementary Data 2), because glutamate as penultimate amino acid was previously shown to increase protein stability in chloroplasts[85]. The DNA fragment was inserted into the chloroplast transformation vectors (pDK323 and pMPH36) by EcoRV linearization and In-Fusion® HD homology-based integration, resulting in SUMO expression vectors pMPH50 and pMPH71. PCR products encoding the proteins to be SUMO tagged and a short sequence restoring the codons for the last two amino acids of

SUMO were inserted into the linearized transformation vectors by In-Fusion® HD homology-based integration.

To obtain SUMO-GFP fusions, the synthetic green fluorescent protein (sGFP) was used[86]. The sGFP DNA sequence used is identical to NCBI GenBank entry DQ370424.1 (NCBI protein ID GenBank: ABD16415.1), but lacks the codons encoding the last two amino acids (I and L), and ends with the stop codon TAA.

## Chloroplast transformation

Vectors for chloroplast transformation were amplified in *E. coli* strain TOP 10, and plasmid DNA was purified with anion exchange columns (Macherey-Nagel). The plasmid preparations were resequenced prior to plant transformation to confirm absence of mutations from the transgenes or the flanking chloroplast homology regions.

Samples of 30 μg vector DNA were coated onto gold particles and shot by particle bombardment into young *Nicotiana tabacum* cv. Petit Havana leaves harvested from plants cultivated under sterile conditions. Bombarded leaves were cut into pieces and incubated on regeneration medium for tobacco shoot organogenesis composed of MS elements + modified vitamins (Duchefa M0245), sucrose (30 g/L), sorbitol (18.22 g/L), mannitol (18.22 g/L), 6-benzylaminopurine (BAP; 1 mg/L), 1-naphthaleneacetic acid (NAA; 0.1 mg/L), agar (5.4 g/L) at pH 5.8, and supplemented with 500 mg/mL spectinomycin (Duchefa). Primary spectinomycin-resistant shoots were subjected to additional regeneration rounds to select for homoplasmy. To this end, callus or leaf pieces of antibiotic-resistant lines were transferred to fresh regeneration medium with 500 mg/mL spectinomycin. In parallel, tissue samples were incubated on regeneration medium containing 500 mg/mL streptomycin (Duchefa), to distinguish true transplastomic clones from spontaneous spectinomycin-resistant mutants[67]. Finally, emerging homoplasmic shoots were rooted on Murashige & Skoog (MS) medium[87] with sucrose (3%) and 500 mg/mL spectinomycin, transferred to soil, and grown to maturity. Per construct, at least three independent transplastomic lines were created.

## DNA gel blot analysis

The complete replacement of the wild-type plastid genome with the transgenic genome (homoplasmy) was confirmed by restriction fragment length polymorphism analysis (RFLP) via Southern blotting. Briefly, DNA was isolated from snap-frozen leaf material by a cetyltrimethylammoniumbromide (CTAB)-based DNA extraction method[88]. DNA concentrations were determined with a Thermo Scientific™ NanoDrop™ spectrophotometer. Samples of 3 μg DNA were digested in a 50 μL volume overnight at 37 °C with the restriction enzyme BglII (NEB). The digest was subsequently mixed with 10 μL 6x DNA gel loading dye (Thermo Fisher Scientific). Samples of 30 μL were electrophoretically separated in a 0.8% agarose gel. The gel was incubated for 15 min in depurination buffer (0.25 M HCl), rinsed with ddH2O and incubated with 0.5 M NaOH for 30 min, rinsed with ddH2O and incubated in 0.5 M NaOH and 1.5 M NaCl for 30 min, and finally rinsed with ddH2O and incubated in 1 M Tris, 3 M NaCl for 15 min. DNA gels were blotted with 10 x SSC buffer (1.5 M NaCl, 0.15 M trisodium citrate) onto a Hybond-XL membrane (GE Healthcare), followed by UV crosslinking (0.12 J/cm$^2$; UV-crosslinker BLX-254, Vilber Lourmat). Restriction fragments were detected by hybridization with a [α-$^{32}$P]-dCTP-labelled *psaB* probe generated by PCR (primers: CCCAGAAAGAGGCTGGCCC and CCCAAGGGGCGGGAACTGC). Overnight hybridization was performed in Church buffer [1% (w/v) BSA, 0.5 M Na2HPO4, pH 7.2, 7% (w/v) SDS, 1 mM EDTA, pH 8] at 65 °C. Membranes were briefly rinsed and incubated with 2 x SSC and 0.1% SDS at 25 °C for 20 min and incubated twice with 0.5 x SSC and 0.1% SDS at 65 °C for 15 min. Membranes were exposed to a storage phosphor screen (GE Healthcare) followed by imaging of the screen with a Typhoon™ TRIO + scanner (GE Healthcare). For images of blots, see Supplementary Figs. 1–3, and for uncropped blots, see Source Data file.

## RNA gel blot analyses

Transgene expression in transplastomic lines was analyzed by northern blotting. Total plant RNA was extracted using TRIzol® (Thermo Fisher Scientific) following the protocol of the manufacturer, and quantified by optical density measurements using a Thermo Scientific™ NanoDrop™ spectrophotometer. Samples of 3 μg total RNA were denatured at 75 °C for 15 min and then separated by agarose gel electrophoresis. Denaturing 1% (w/v) agarose gels were prepared in 1x MOPS buffer [0.1 M 3-(N-morpholino) propanesulfonic acid, 0.3 M NaOAc, 1 mM EDTA; pH 7] in presence of 16% (v/v) formaldehyde (Sigma). Separated RNA was transferred onto Hybond-N nylon membranes (GE Healthcare) by overnight capillary blotting using 5 x SSC buffer (0.75 M NaCl, 75 mM trisodium citrate; pH 7), and covalently bound to the membrane by exposure to UV light (0.12 J/cm$^2$; UV-crosslinker BLX-254, Vilber Lourmat). Blotted nylon membranes were stained with 0.03% methylene blue (SERVA) solution in 0.3 M NaOAc, pH 5.2, and scanned using an EPSON Perfection V700 Photo scanner. A SpeI/EcoRV restriction fragment digested from vector pDK323 (corresponding to the *Chlamydomonas reinhardtii atpA* 3′ UTR) was labelled with [α-$^{32}$P]-dCTPs (Hartmann Analytic GmbH) using the Amersham™ Megaprime™ DNA Labeling System (Cytiva) following the manufacturer's protocol. Hybridization was performed overnight at 65 °C in Church buffer [1% (w/v) BSA, 0.5 M Na2HPO4 (pH 7.2), 7% (w/v) SDS, 1 mM EDTA (pH 8.0)]. Membranes were exposed to a storage phosphor screen (GE Healthcare) followed by radioactive signal detection using the Typhoon™ TRIO + scanner (GE Healthcare). For images of uncropped blots, see Source Data file.

## Plant growth under sterile conditions

Plants were grown under aseptic conditions in plastic boxes (Magenta) on MS medium with 3% sucrose (with or without 500 mg/L spectinomycin) under a diurnal cycle of 50 μmol m$^{-2}$ s$^{-1}$ light for 16 h at 25 °C, and 8 h darkness at 20 °C. Plants that did not produce seeds in the greenhouse were maintained in sterile culture by rooting of stem cuttings every three to six months. Pigment-deficient plants were maintained at minimal light intensities of 3–5 μmol m$^{-2}$ s$^{-1}$.

## Inheritance assays

To confirm the homoplasmic state of transplastomic plants, surface sterilized seeds were germinated on antibiotic-containing MS medium with 3% sucrose. Antibiotic-resistant seedlings were identified by selection in the presence of spectinomycin (500 μg/mL).

## Greenhouse growth of plants

Tobacco plants (*Nicotiana tabacum* var. Petit Havana) were grown in soil in the greenhouse with light supplementation. Artificial light was used from 6 am to 10 pm (i.e., for 16 h). The measured total average light intensity was (PPFD)/(PAR) 195 μmol m$^{-2}$ s$^{-1}$, the day temperature set to 25 °C, the night temperature 20 °C, and the relative humidity set to 55%.

## Grafting

To allow growth on soil and seed production of AMP-expressing plants that displayed severe pigment-deficient phenotypes and did not grow autotrophically, transplastomic shoots were grafted onto wild-type shoots in tissue culture. Autoclaved pieces of silicon tubing, cut open longitudinally to allow later removal, were used to join wild-type stock and transplastomic scion. Successful grafts were transferred to soil and grown in the greenhouse. Wild-type side shoots were pruned every 14 to 30 days to prevent overgrowth and encourage growth and flowering of the mutant shoot.

## Total protein extraction

To extract total leaf protein from tobacco plants, leaf pieces were snap frozen in liquid nitrogen in 2 mL Eppendorf tubes and pulverized with 5 mm steel balls for one minute and 15 s at 20 Hz in a Retsch mill. Total

protein was isolated by a phenol-based extraction protocol[89]. Protein pellets were dissolved in 1% sodium dodecyl sulfate (SDS). Protein concentrations were quantified with the Pierce BCA Protein Assay Kit (Thermo Fisher Scientific) and measured with a CLARIOstar plate reader (BMG Labtech) using a bovine serum albumin (BSA) dilution series as standard. Calculations of protein concentrations based on spectrophotometric data were done with Microsoft Excel (Microsoft 365 version 2202).

## His-tag-based protein purification

Leaf material was snap frozen in liquid nitrogen, ground with a Retsch mill in metal containers for one minute at 25 Hz, and incubated with extraction buffer[24] (100 mM Tris, 300 mM NaCl, 20 mM ascorbic acid, 10 mM sodium metabisulfite, 10 mM imidazole, final pH 8.0). 100 μL buffer was used per 100 mg leaf material, and the samples were incubated on ice for approximately 30 min with regular mixing by vortexing. Insoluble material was pelleted by centrifugation (16,000 x $g$ for 30 min at 4 °C). The supernatant was used as input for purification. For large-scale purification, frozen leaf material was ground with buffer (3 mL per g leaf material) in a countertop blender (two short pulses of 10 s at low speed, followed by three rounds of grinding for 60 s at high speed). Tissue debris and other insoluble material were pelleted in large beakers by centrifugation (15,000 x $g$ for 30 min at 4 °C). The supernatant (lysate) was decanted through filter papers, and the resulting filtrate used as input for purification.

Nickel-nitrilotriacetic acid (Ni-NTA) agarose beads (QIAGEN) were resuspended to a slurry in the storage solution. The slurry volume required for purification was transferred into an Eppendorf tube, the beads were pelleted by centrifugation and the supernatant discarded. The beads were then equilibrated by washing once with 2.5 volumes of ddH$_2$O and twice with 2.5 volumes of extraction buffer. Pelleting by centrifugation was performed at 700 x $g$ for 10–15 min at 4 °C. For small batch purification, bead slurry equivalent to 10% or 60% of the volume of the protein extract was used for subsequent protein purifications.

For purification, beads were incubated with protein extract by slow rotation for one hour, washed twice with 2.5 slurry volumes of washing buffer (extraction buffer plus 20 mM imidazole), and eluted with 25 or 33% of the protein extract input volume of elution buffer (extraction buffer with 250 mM imidazole). All purification steps were performed at 4 °C.

For large-scale batch purifications, equilibrated Ni-NTA agarose beads (2% of the lysate volume) were incubated in the supernatant with a magnetic stirrer bar for one hour on ice. Subsequently, the lysate with the beads was decanted in several batches onto gravity columns (QIAGEN 5 mL polypropylene columns), allowing the Ni-NTA agarose beads to sediment on the filter while letting the lysate to flow through. The columns were then washed 3 times with 10 mL of washing buffer (for higher purity) or extraction buffer (for higher yield), and eluted 4 times with 1 mL of elution buffer.

Eluted fractions (3.5 mL of 4 mL) were loaded on a PD-10 Desalting Column (GE Healthcare), starting with the most concentrated fractions, buffer-exchanged into 20 mM Tris/HCl pH 8.0, and concentrated by ultrafiltration (Amicon® Ultra-4 centrifugal filters; Merck) with a 3000 Nominal Molecular Weight Limit (NMWL) cut-off. Protein concentrations were determined with the Pierce BCA Protein Assay Kit (Thermo Fisher Scientific) and measured with a CLARIOstar plate reader (BMG Labtech) using a dilution series of BSA as standard. Protein concentrations based on spectrophotometric data were calculated with Microsoft Excel (Microsoft 365 version 2202).

## Tris-tricine SDS-PAGE and immunoblotting

To analyze protein accumulation and follow protein purification, protein extracts were separated by electrophoresis in 10% Tris-tricine SDS-polyacrylamide gels[90,91]. Protein extracts were mixed with equal volumes of 2 x Tris-tricine sample buffer (24% glycerin, 0.1 M Tris pH

6.8, 8% SDS, 0.2 M DTT, 0.02% Coomassie Brilliant Blue G-250), denatured for 5 min at 95 °C, centrifuged for 1 min at 15,000 x $g$ and then loaded onto the gel. Gels were run overnight at 16 °C at constant voltage.

For western blotting, protein gels were electroblotted to Hybond-P PVDF membranes (GE Healthcare) in transfer buffer (192 mM glycine, 25 mM Tris, pH 8.3) in a Trans-Blot cell (Bio-Rad) for 2 h at 1.0 A at 16 °C.

Blotted membranes were stained with Coomassie prior to immunochemical detection. To this end, the membranes were incubated with Coomassie 250 R (50 mg per 200 mL in 50% methanol and 7% acetic acid) directly after transfer. Membranes were stained for 2–5 min until the membrane was dark blue, and subsequently destained with 50% methanol and 7% acetic acid for one hour or until the background was sufficiently reduced. The membranes were then rinsed briefly with ddH$_2$O and scanned. The Coomassie was subsequently removed with 100% methanol by incubating for up to one hour, and the membranes were washed with Tris-Buffered Saline (TBS) with 1% TWEEN® 20 (v/v; TBS-T) before proceeding with the blocking procedure prior to immuno-detection.

Blotted membranes were blocked with 5% milk powder (w/v) in TBS-T for one hour at room temperature. The membranes were then rinsed twice with TBS-T and subsequently washed once for 15 min and twice for 5 min in TBS-T on a shaker. The blots were subsequently incubated with the primary antibody (anti-HA; GenScript THE™ HA-tag Antibody, mAb, mouse, catalogue No: A01244-100; 0.5 μg/mL) for one hour at room temperature or overnight at 4 °C. Primary antibodies were prepared in TBS-T with 200 μL Micro-O-Protect (80% ethanol, 7.5% 5-bromo-5-nitro-1,3-dioxane, 7.5% 2-methyl-4-isothiazolin-3-one hydrochloride) and stored at 4 °C. After incubation with the antibody solution, the membranes were washed as described above, and incubated with HRP-conjugated anti-mouse antibody (Sigma A9044; diluted freshly in TBS-T 1:10.000). Afterwards, the membranes were washed again, and the signals were detected with the enhanced chemiluminescence kit (ECL® PLUS system, GE Healthcare). Images of uncropped blots are available in the Source Data file.

## Antimicrobial activity assays in vitro

A modified radial diffusion assay[92–94] was used to test antimicrobial activity of protein extracts. *E. coli* TOP 10 cells were inoculated overnight at 37 °C in liquid casein peptone soybean flour peptone (CASO) medium. 10 μL samples of the overnight cultures were inoculated in 10 mL CASO, and grown for 3 h. The cultures were then pelleted by centrifugation (3000 x $g$, 10 min, 20 °C) and the pellets were rinsed carefully with cold (4 °C) 10 mM sodium phosphate buffer, pH 7.4 (NAPB), followed by resuspension in 5 mL NAPB. The OD$_{620}$ was measured in the suspension and a 10x diluted suspension, and the number of colony-forming units (CFU) was determined by the following equation 1.

$$CFU/mL = OD_{620} \times 2.5 \times 10^8 \qquad (1)$$

The volume of bacterial suspension containing $4 \times 10^6$ CFU was calculated and added to 10 mL of melted (55 °C) bottom layer medium. The liquid bottom layer medium was mixed with the bacteria by quickly inverting the tube 10 times, and poured into a Petri dish (8.5 cm diameter). The bottom layer medium was composed of 10 mM sodium phosphate buffer (NAPB), pH 7.4, 100x diluted CASO liquid medium, and 1% agarose (for nucleic acid electrophoresis, Sigma).

Protein samples to be assayed were loaded into wells in the bottom layer. Wells were made with 3 mL syringes (Braun) by pressing the tip (without an attached needle) into the agar, bringing up the plunger slightly and pulling the syringe up quickly. The resulting wells were approximately 4 mm in diameter. Samples (of 10 μL) were loaded into the wells in the agarose, and allowed to diffuse for two to three hours at 37 °C. Subsequently, 10 mL nutritious top layer (2x CASO medium

with 1% agarose) was poured on top of the bottom layer. After solidification of the top layer, plates were incubated upside down overnight at 37 °C, and inspected the next morning for zones without bacterial growth surrounding the wells.

### SUMO digestion

SUMO protease digestion was performed according to a published protocol[76] with some modifications. SUMO protease (2500 units 6xHis-tagged Ulp1; Sigma Aldrich, SAE0067-2500UN) was diluted to 25 units per μL in 50% glycerol with 1 mM DTT, aliquots were snap frozen in liquid nitrogen and stored at −20 °C until use. Digestion was performed in digestion buffer (20 mM Tris/HCl, pH 8.0) for 2–3 h at room temperature (25 °C). SUMO digestion of fusion proteins was performed with a final concentration of 0.25 to 0.83 units per μL, and a total amount of 2.5 to 25 units per 25 to 170 μg of protein substrate. Cleavage efficiency was determined by image quantification of western blots with ImageJ version v1.52r (https://imagej.nih.gov/ij/).

### Reporting summary

Further information on research design is available in the Nature Research Reporting Summary linked to this article.

## Data availability

A reporting summary for this Article is available as a Supplementary Information file. Data supporting the findings of this work are available within the paper and its Supplementary Information files. Source data are provided with this paper.

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

## Acknowledgements

We thank Dr. Stephanie Ruf and her team (in particular Jennifer Bergander and Luisa Heinig) for help with plant transformation and tissue culture, the MPI-MP GreenTeam for plant cultivation, the MPI-MP photographers for taking pictures of the plants, and Prof. Dr. Dirk Walther for help with codon optimization of the linker sequences. This research project received funding from the Max Planck Society and the European Research Council (ERC) under the European Union's Horizon 2020 research and innovation programme (ERC-ADG-2014; grant agreement No 669982) to R.B.

## Author contributions

M.P.H., J.F. and R.B. designed the experiments, M.P.H. performed the majority of the experiments, S.C. contributed to the development and characterization of the SUMO tag system, C.K. and Z.T. contributed to the molecular characterization of transplastomic plants, F.V.L. performed northern blots depicted in this article, S.A. and D.K. developed the split transformation system with RT7RP and PT7, D.K. also created expression vector pDK323, F.M. contributed to DNA construct generation and development of protein purification methods, X.K. performed a major part of the tissue culture and plant transformation experiments, all authors interpreted and analyzed data, R.B. and M.P.H. wrote the paper with input from all other authors.

## Funding

## Competing interests
The authors declare no competing interests.
