## [Peer Review File · Nature Communications]

Expression strategies for the efficient synthesis of antimicrobial peptides in plastidsReviewers' Comments:

Reviewer #1:

Remarks to the Author:

In this work, Hoelscher and coworkers described their attempt to express antimicrobial peptides in chloroplasts after integration of recombinant genes encoding the AMPs into chloroplast genomes. Authors have examined various strategies in designing the recombinant genes encoding AMPs regarding to the copy number, the length of linkers, inducible expression, etc. After generating the transgenic plants harboring these recombinant genes in the chloroplast genomes, authors examined the phenotype, growth behavior and expression of their AMP coding genes.

In fact, authors generated very large number of transgenic plants and examined the expression of the AMPs in plants. As expected, a large proportion of transgenic plants showed a varying degree of problems in growth and greening. Eventually authors found that SUMO-fused AMP appears to be most successful in reducing toxicity, thereby producing more wild-type looking plants. Also authors purified the AMPs and found that SUMO-fused AMPs exhibited activity against bacteria independently of proteolytic removal of the carrier protein.

This manuscript contains biotech-related work. Therefore, the main focus should be the expression level and functional activity of AMPs. The AMPs are active when they are produced, which is very much expected. However, the expression level was not very much impressive. This reviewer expects that some good expression vectors used in transient expression would give similar yield. Thus this reviewer does not see much novelty in this work to be deserved in publication in Nature Comms.

Reviewer #2:

Remarks to the Author:

Expression strategies for the efficient synthesis of antimicrobial peptides in plastids, by Hoelscher et al.

In this manuscript, Hoelscher et al. describe expression strategies for different antimicrobial peptides (AMPs) in tobacco plasmids. The production of AMPs in plants is an interesting topic, and so far several labs have addressed this problem. Unfortunately, the presented manuscript needs a lot of additional data before it should be accepted for publication in Nat Communications.

Actually, the manuscript describes and addresses what is promised in the title: How can peptides be produced in plastids?

My problems:

1. The manuscript does not (!) address whether the peptides are active.
2. It does not show whether the peptides can be cleaved from the SUMO construct.
3. It does not give the MIC50 for the expressed and purified peptides.
4. The group of Ralf Bock has published expression of AMPs in plastids. What is so new and exciting to submit the story to Nat Communications?

Please address these points

Reviewer #3:

Remarks to the Author:

The present manuscript addresses the issue of antimicrobial peptides (AMPs) that are produced by various organisms and may represent a new way to cope with increasing bacterial resistance to conventional antibiotics. One of the significant obstacles to using AMPs is their availability, as chemical synthesis, if feasible, is very expensive and recombinant production in the bacterial host has many limitations. The authors present a study on the production of AMPs in tobacco plants, using an integration of expression cassette into the chloroplast genome. Several major challenges had to be overcome to make this expression possible, such as stability of the protein products towards degradation by endogenous proteases and toxicity of the products to the plant host. Under constitutive expression, albino transplastomic plants incapable of normal growth were obtained, which was partially overcome by inducible expression and further improved by SUMO fusion of the product. The study is innovative and deserves publication, but some points should be addressed:

- There have been previous studies on recombinant production of AMPs in plants, using the integration of the expression cassette into the nuclear genome by using either constitutive or tissue-specific promoters. A short comment on these studies and representative citations should be included in the Introduction.
- L80: The reference 12 is somewhat outdated (2005) and should be replaced with a more recent one.
- L97-101: These sentences are misleading and should be rewritten. Small plastid targeting peptides usually do not carry a positive charge; they are often rich in alanine, serine and other polar, but uncharged amino acids.
- There is a quantification of the obtained recombinant protein products presented on L387-392 but no indication of how these values were obtained. The method of determining the yield should be described along with the protein standard used for the calculation and appropriate statistics.
- The yields should be expressed in the amount of the AMP itself, not a fusion with several domains.
- The cleavage of the fusion proteins with SUMO protease seems to be incomplete (Fig. 7 b). An explanation and an estimated ratio of the cleaved/uncleaved product should be provided.
- In general, chloroplast expression of transgenes is supposed to be superior to other means of production of recombinant proteins in plants. The obtained AMP yields should be discussed in relation to other works on AMPs or similar products in plants.

Reviewer #4:

Remarks to the Author:

The rise of antibiotic resistance in pathogenic bacteria presents a major challenge to human health. AMPs provide a potential solution to treat infections resistant to conventional antibiotics. This is limited by their high cost, which has been addressed by using a low cost production platform in this manuscript. The manuscript contains a comprehensive set of results, which are well organised. The use of SUMO fusions combined with inducible expression, was required to overcome the toxic properties of the AMPs. The results are a significant breakthrough and should be of widespread interest. High-yield chloroplast expression of AMPs (as reported in this work) provides a new route for manufacturing AMPs to combat diseases caused by bacteria resistant to conventional antibiotics.

Other comments

1. Abstract. The abstract could mention the total number of AMPs tested singly and as multimers, and capture the variation in response to fusing with SUMO, with some but not all requiring a combination of inducible expression and SUMO fusion. The yield of purified product could be included, which

appears to be 40-55 mg per kg of leaves.

2. Introduction lines 75-84 and Discussion. If known, the cost range of chloroplast expressed AMPs versus the costs of chemical synthesis would be of interest to the general reader.

3. Introduction lines 98-109. The text provides an interesting comparison between transit peptide and the sizes of recombinant protein. A comment on the smallest detectable chloroplast encoded protein might be relevant here.

4. line 250. fAMPs ((f)AMPs). Why is '(f)AMPs' included in brackets

5. line 255. 'Hybridization'. Antibodies bind rather than 'hybridize'

6. line 340 and 344. The sentences appear to contradict each other. 'Fusing 3-Wamdeposin-5 to SUMO reduced the toxicity of this polypeptide to the plant.' but this statement appears to be contradictory to , 'Prn::SUMO-3-Wamdeposin-5 plants were completely white, resembling the phenotype of Prn::3-Wamdeposin-5 plants.'

7. Fig 6c. Lower panel. The SUMO-GFP fusion is clearly visible by staining the membrane. Are any of the SUMO-AMP fusions visible on a Coomassie BB stained gel or membrane? The lower section of this stained membrane would show this. Could this be included?

8. Line 1160. The protein is rubisco LS. rbcL is the gene.

9. Fig 7a The Sumo-AMP band in the In (supernatant) lanes could be marked with a dot or arrow.

10. Line 376. 'Protein purification was performed successfully for SUMO-HA-WAM1, SUMO-HA-Novispirin and SUMO-GFP using buffers without detergents.' However, the text and Fig 7a indicates limited purification of SUMO-3-Wamdeposin-5, which is not consistent with successful purification.

11. line 447. 'Third, fusion to SUMO also alleviated the mutant phenotypes associated with high-level fAMP expression.'

This was not universal as stated in sentence (line 345) 'By contrast, Prn::SUMO-3-Wamdeposin-5 plants were completely white, resembling the phenotype of Prn::3-Wamdeposin-5 plants.'

12. As demonstrated by the authors, the SUMO polypeptide can be cleaved off the purified protein using the protease If relevant applications requiring removal of SUMO could be mentioned.

13. Two of the sumo AMP fusions were tested on E. coli TOP10 cells (Fig 7c) and the extent to which this bactericidal activity would apply to gram negative or gram positive pathogens might be mentioned. Any data on the MICs of the fAMPs would be of interest although this may be outside the scope of the present work which was to develop an expression system capable of producing large amounts of AMPs and fAMPs.

14. Lines 327-330. Indicate the size of the SUMO polypeptide in amino acids and kDa.

REVIEWER COMMENTS (with our response in blue)

Reviewer #1 (Remarks to the Author):

In this work, Hoelscher and coworkers described their attempt to express antimicrobial peptides in chloroplasts after integration of recombinant genes encoding the AMPs into chloroplast genomes. Authors have examined various strategies in designing the recombinant genes encoding AMPs regarding to the copy number, the length of linkers, inducible expression, etc. After generating the transgenic plants harboring these recombinant genes in the chloroplast genomes, authors examined the phenotype, growth behavior and expression of their AMP coding genes.

In fact, authors generated very large number of transgenic plants and examined the expression of the AMPs in plants. As expected, a large proportion of transgenic plants showed a varying degree of problems in growth and greening. Eventually authors found that SUMO-fused AMP appears to be most successful in reducing toxicity, thereby producing more wild-type looking plants. Also authors purified the AMPs and found that SUMO-fused AMPs exhibited activity against bacteria independently of proteolytic removal of the carrier protein.

This manuscript contains biotech-related work. Therefore, the main focus should be the expression level and functional activity of AMPs. The AMPs are active when they are produced, which is very much expected. However, the expression level was not very much impressive. This reviewer expects that some good expression vectors used in transient expression would give similar yield. Thus this reviewer does not see much novelty in this work to be deserved in publication in Nature Comms.

We respectfully disagree with the assessment that our AMP expression levels are unimpressive, because similar yields might be expected with transient expression. First, AMP expression has been challenging in all systems that have been tried so far. Second, when judging accumulation levels, it needs to be considered that AMPs are very small polypeptides (and even our fusions are still small!). Consequently, even moderate protein accumulation levels (e.g., of 56 μg SUMO-AMP per g leaf material) correspond to enormous molar concentrations of the AMPs and impressive yields. Third, our stably transformed transplastomic plants offer significant advantage over transient expression systems (even if transient expression would produce similar AMP levels), including absence of transgenic pathogenic microbes (*Agrobacterium* or viruses), lack of the requirement for transformation of each new batch of plant material (also resulting in much greater batch-to-batch consistency in protein accumulation levels), and absence of epigenetic transgene silencing. We, therefore, believe that the successful development of AMP expression strategies based on stable plastid genome transformation and their thorough optimization provide important new insights in the properties and stability of AMPs, and in addition, provide a novel cost-efficient production platform for these important therapeutic molecules. We feel that these are very substantial achievements that deserve publication in *Nature Communications*. To make the attractions of the chloroplast system clearer, we have revised the Introduction (p. 4 & 5) and Discussion (p. 22) sections, and included a comparison of different expression systems (including a published report on virus-based transient expression of a cationic AMP).

Reviewer #2 (Remarks to the Author):

Expression strategies for the efficient synthesis of antimicrobial peptides in plastids, by Hoelscher et al.

In this manuscript, Hoelscher et al. describe expression strategies for different antimicrobial peptides (AMPs) in tobacco plasmids. The production of AMPs in plants is an interesting topic, and so far several labs have addressed this problem. Unfortunately, the presented manuscript needs a lot of additional data before it should be accepted for publication in Nat Communications.

Actually, the manuscript describes and addresses what is promoted in the title: How can peptides be produced in plastids?

My problems:

1. The manuscript does not (!) address whether the peptides are active.

We have demonstrated antimicrobial activity for two of the produced AMPs and the data are shown in Figure 7c. This finding is relatively unsurprising, because due to their small size and their generally low folding requirements, AMPs are active even when chemically synthesized.

2. It does not show whether the peptides can be cleaved from the SUMO construct.

Cleavage of two AMPs from the SUMO construct is shown in Figure 7b of the revised manuscript.

3. It does not give the MIC50 for the expressed and purified peptides.

See above (point 1.). The main focus of our work has been to develop solutions for the big challenges associated with the synthesis of AMPs (and other small polypeptides) in plastids. We focused less on the characterization of the purified products (for the reason stated under 1.), and believe that our data illustrate the great potential of our developed methods for recombinant production of valuable peptides in plastids.

4. The group of Ralf Bock has published expression of AMPs in plastids. What is so new and exciting to submit the story to Nat Communications?

The Reviewer correctly points out that some antimicrobial agents have been produced by us previously. However, these were very large proteins (so-called endolysins) and not antimicrobial peptides (AMPs). They have no stability problem and can be readily expressed in the chloroplast. However, they are less useful as therapeutic agents, because they are large enzymes that have limited applications in that their administration in medicine will cause strong undesired immune responses. These key differences are now properly explained (and referenced) in the Discussion section of our revised manuscript (p. 20).

Reviewer #3 (Remarks to the Author):

The present manuscript addresses the issue of antimicrobial peptides (AMPs) that are produced by various organisms and may represent a new way to cope with increasing bacterial resistance to conventional antibiotics. One of the significant obstacles to using AMPs is their availability, as chemical synthesis, if feasible, is very expensive and recombinant production in the bacterial host has many limitations. The authors present a study on the production of AMPs in tobacco plants, using an integration of expression cassette into the chloroplast genome. Several major challenges had to be overcome to make this expression possible, such as stability of the protein products towards degradation by endogenous proteases and toxicity of the products to the plant host. Under constitutive expression, albino transplastomic plants incapable of normal growth

were obtained, which was partially overcome by inducible expression and further improved by SUMO fusion of the product. The study is innovative and deserves publication, but some points should be addressed:

1- There have been previous studies on recombinant production of AMPs in plants, using the integration of the expression cassette into the nuclear genome by using either constitutive or tissue-specific promoters. A short comment on these studies and representative citations should be included in the Introduction.

As suggested by the Reviewer, we have added to the Introduction mentioning the different expression systems and citing previous work on constitutive and transient expression of AMPs and other recombinant polypeptides in plants (p. 4 & 5).

2- L80: The reference 12 is somewhat outdated (2005) and should be replaced with a more recent one. ¹

We agree and have added a more recent reference (p. 4).

3- L97-101: These sentences are misleading and should be rewritten. Small plastid targeting peptides usually do not carry a positive charge; they are often rich in alanine, serine and other polar, but uncharged amino acids.

The Reviewer is correct: positive charges are present in mitochondrial transit peptides, but not so much in chloroplast transit peptides. We have removed the statement and instead, included an extra sentence about the substrate preferences of PreP (that is present in both mitochondria and plastids) for charged residues next to the cut site (p. 5 & 6).

4- There is a quantification of the obtained recombinant protein products presented on L387-392 but no indication of how these values were obtained. The method of determining the yield should be described along with the protein standard used for the calculation and appropriate statistics.

We apologize for this omission, and have added the information to the Methods section (p. 32). We also added the raw data of the concentration measurements and the calculations behind the reported values to the supplementary Source Data file.

5- The yields should be expressed in the amount of the AMP itself, not a fusion with several domains.

Based on the molecular weight of SUMO and that of the fused AMP (or GFP), we calculated the AMP and GFP yields, as suggested by the Reviewer. This information is now provided in the revised manuscript (p. 18) and the underlying calculations are included in the Source Data file.

6- The cleavage of the fusion proteins with SUMO protease seems to be incomplete (Fig. 7 b). An explanation and an estimated ratio of the cleaved/uncleaved product should be provided.

As suggested by the Reviewer, we have determined the cleavage efficiency and found it to be approximately 80% for SUMO-GFP and SUMO-Novispirin, and 60% for SUMO-HA-WAM1. This has been added to the Results (p. 18) and the Methods (p. 35), and a likely reason for the incomplete cleavage (presence of a subfraction of the purified fusion protein in a conformation that is not accessible by the protease) is briefly mentioned in the Discussion section (p. 20). The image quantifications and calculations to determine cleavage efficiency have been added to the Source Data file.

7- In general, chloroplast expression of transgenes is supposed to be superior to other means of production of recombinant proteins in plants. The obtained AMP yields should be discussed in relation to other works on AMPs or similar products in plants.

We have added a paragraph (p. 22) to the Discussion summarizing previous attempts to express AMPs in plants, also covering anionic AMPs (which are easier to express, but much less potent as therapeutic agents).

Reviewer #4 (Remarks to the Author):

The rise of antibiotic resistance in pathogenic bacteria presents a major challenge to human health. AMPs provide a potential solution to treat infections resistant to conventional antibiotics. This is limited by their high cost, which has been addressed by using a low-cost production platform in this manuscript. The manuscript contains a comprehensive set of results, which are well organized. The use of SUMO fusions combined with inducible expression, was required to overcome the toxic properties of the AMPs. The results are a significant breakthrough and should be of widespread interest. High-yield chloroplast expression of AMPs (as reported in this work) provides a new route for manufacturing AMPs to combat diseases caused by bacteria resistant to conventional antibiotics.

Other comments

1. Abstract. The abstract could mention the total number of AMPs tested singly and as multimers, and capture the variation in response to fusing with SUMO, with some but not all requiring a combination of inducible expression and SUMO fusion. The yield of purified product could be included, which appears to be 40-55 mg per kg of leaves.

As suggested, we have added the total number of tested AMPs and independent sets of transplastomic plants to the Abstract (and removed some other words, to stay within the word limit of 150 words allowed), but did not manage to create space for inclusion of more detail.

2. Introduction lines 75-84 and Discussion. If known, the cost range of chloroplast expressed AMPs versus the costs of chemical synthesis would be of interest to the general reader.

The cost of commercial-scale chemical synthesis are difficult to determine, not the least because they strongly depend on polypeptide length and the requirements for post-translational modifications (for example, many AMPs require disulfide bridges). To take this into account, we have modified the sentence in the Introduction as follows: ‘... expensive (with the possible exception of some very small AMPs that do not require disulfide bridges or other post-translational modifications).’ (p. 3 & 4).

3. Introduction lines 98-109. The text provides an interesting comparison between transit peptide and the sizes of recombinant protein. A comment on the smallest detectable chloroplast encoded protein might be relevant here.

This is an excellent suggestion. We have added a paragraph to the Introduction (p. 6) explaining how small chloroplast-encoded proteins escape proteolytic degradation (all of them are subunits of large multiprotein complexes, and their rapid incorporation into the complexes protects them from degradation).

4. line 250. fAMPs ((f)AMPs). Why is ‘((f)AMPs)’ included in brackets

We used the term (f)AMP to refer to both unfused and fused AMPs, but realized that we had not properly explained that. This is now rectified by addition of the phrase ‘AMPs and fAMPs (hereafter collectively referred to as (f)AMPs)’ (p. 12).

5. line 255. ‘Hybridization’. Antibodies bind rather than ‘hybridize’

We agree and have removed the word ‘hybridization’ (and replaced it by ‘recognized by the antibody’) (p. 19) or deleted the word (p. 12).

6. line 340 and 344. The sentences appear to contradict each other. ‘Fusing 3-Wamdeposin-5 to SUMO reduced the toxicity of this polypeptide to the plant.’ but this statement appears to be contradictory to , ‘Prn::SUMO-3-Wamdeposin-5 plants were completely white, resembling the phenotype of Prn::3-Wamdeposin-5 plants.’

We have modified this sentence to make clear that SUMO fusion alleviates the strong mutant phenotypes in constructs resulting in low to moderate expression levels, but not in the very strong constitutive Prn::SUMO-3-Wamdeposin-5 plants (p. 16).

7. Fig 6c. Lower panel. The SUMO-GFP fusion is clearly visible by staining the membrane. Are any of the SUMO-AMP fusions visible on a Coomassie BB stained gel or membrane? The lower section of this stained membrane would show this. Could this be included?

Some SUMO::AMP fusions are clearly visible in Coomassie-stained blots. They are now indicated by asterisks in the total protein fractions in Fig. 7a. In addition, uncropped images of all stained membranes have been included in the Source data File.

8. Line 1160. The protein is rubisco LS. rbcL is the gene.

RbcL has been replaced by ‘large subunit of RuBisCO’ throughout the manuscript.

9. Fig 7a The Sumo-AMP band in the In (supernatant) lanes could be marked with a dot or arrow.

Done.

10. Line 376. ‘Protein purification was performed successfully for SUMO-HA-WAM1, SUMO-HA-Novispirin and SUMO-GFP using buffers without detergents.’ However, the text and Fig 7a indicates limited purification of SUMO-3-Wamdeposin-5, which is not consistent with successful purification.

We agree and have removed the sentence about SUMO-3-Wamdeposin-5 (p. 17). (The limited purification efficiency is likely due to the 3-Wamdeposin-5 being a larger hydrophobic cationic polypeptide in comparison to the single AMPs). In addition, plants producing detectable levels of SUMO-3-Wamdeposin-5 could only grow in tissue culture, excluding these plants from real life application.

11. line 447. ‘Third, fusion to SUMO also alleviated the mutant phenotypes associated with high-level fAMP expression.’

This was not universal as stated in sentence (line 345) ‘By contrast, Prn::SUMO-3-Wamdeposin-5 plants were completely white, resembling the phenotype of Prn::3-Wamdeposin-5 plants.’

We agree with the Reviewer, and have modified the sentence as follows: ‘Third, combined with lowered expression levels, fusion to SUMO also alleviated the mutant phenotypes associated with fAMP expression.’ (p. 21).

12. As demonstrated by the authors, the SUMO polypeptide can be cleaved off the purified

protein using the protease. If relevant applications requiring removal of SUMO could be mentioned.

As suggested, we have added two sentences to highlight examples of successful use of cleavable SUMO fusions in bacteria (p. 7 of the revised ms).

13. Two of the sumo AMP fusions were tested on *E. coli* TOP10 cells (Fig 7c) and the extent to which this bactericidal activity would apply to gram negative or gram positive pathogens might be mentioned.

This information has been added to the Discussion section of the revised manuscript (p. 22). We also cite the relevant literature that describes the activities of these AMPs against both gram-negative and gram-positive pathogens.

Any data on the MICs of the fAMPs would be of interest although this may be outside the scope of the present work which was to develop an expression system capable of producing large amounts of AMPs and fAMPs.

This has indeed been outside of the scope of our study, for two reasons. First, the main focus of our work has been to develop solutions for the big challenges associated with the synthesis of AMPs (and other small polypeptides) in plastids. Second, we have demonstrated antimicrobial activity for two of the produced AMPs (Fig. 7). In general, the properties of the AMPs have been well characterized in previous studies, and their activity has been shown to be independent of the synthesis method (e.g., chemical synthesis vs. expression in bacterial cells). This is now mentioned in the Discussion of the revised manuscript (p 22).

14. Lines 327-330. Indicate the size of the SUMO polypeptide in amino acids and kDa. This information has been added (p. 15).

Reviewers' Comments:

Reviewer #2:

Remarks to the Author:

The paper has much improved.

Reviewer #3:

Remarks to the Author:

The authors present the revised study on the production of antimicrobial peptides in the chloroplast genome of tobacco. Although the yields of the peptides are rather low, the study is paving the way to further works leading to application.

All my comments have been all addressed accordingly therefore I recommend the paper for publication.

Reviewer #4:

Remarks to the Author:

The authors have addressed all points raised in my previous review.

REVIEWERS' COMMENTS

Reviewer #2 (Remarks to the Author):

The paper has much improved.

We thank the reviewer for this assessment.

Reviewer #3 (Remarks to the Author):

The authors present the revised study on the production of antimicrobial peptides in the chloroplast genome of tobacco. Although the yields of the peptides are rather low, the study is paving the way to further works leading to application.

All my comments have been all addressed accordingly therefore I recommend the paper for publication.

We thank the reviewer for this assessment.

Reviewer #4 (Remarks to the Author):

The authors have addressed all points raised in my previous review.

We thank the reviewer for this assessment.